# How to Regulate the Infringements of Geographical Indications of Agricultural Products—An Empirical Study on Judicial Documents in China

**DOI:** 10.3390/ijerph20064946

**Published:** 2023-03-11

**Authors:** Lingling Li, Yingzi Chen, Haoran Gao, Changjian Li

**Affiliations:** 1School of Humanities and Social Development, Northwest A & F University, Yangling, Xianyang 712100, China; li_lingling@nwafu.edu.cn (L.L.); chenyingzi@nwafu.edu.cn (Y.C.); gaohr@nwafu.edu.cn (H.G.); 2School of Law, Huazhong University of Science and Technology, Wuhan 430223, China

**Keywords:** geographical indications of agricultural products, infringement, judicial practice, regulation

## Abstract

Under the background of China’s strategy of becoming a powerful agricultural country, geographical indications (GIs) of agricultural products, as an important intellectual property right to enable Chinese agriculture to develop with high quality, have a strong effect of strengthening and promoting agriculture. However, there are a large number of infringements of GIs among agricultural products in judicial practice, which not only greatly damage the economic and social values of GIs of agricultural products, but also bring huge food safety hazards to consumers and hinder the overall protection of intellectual property rights in China. On this basis, this paper, with the help of a quasi-case research method, integrates the facts of relevant cases, the focus of disputes, the application of law, and other case elements to realize the case similarity judgment based on the legal argumentation model. With the help of the retrieval tool of “Peking University Magic Weapon”, this paper provides statistics on the civil cases of infringement of GIs of agricultural products in China from 1 January 2014 to 31 July 2022 and sets different retrieval conditions for two searches. After two screenings, 245 valid samples were obtained, and the judicial patterns of infringement disputes over GIs of agricultural products in China were systematically sorted out from the distribution of plaintiff and defendant, the distribution of infringement types, the basis of adjudication, and the standard of compensation. It was found that the plaintiff types showed double simplification, the infringement types took edge infringement as the basic form, and the general trademark provisions occupied the main position in legal applications. Then, the main litigation points, such as the dispute over the identification of GIs of agricultural products, the dispute over the use of geographical names, and the dispute over tort liability, are summarized, so as to dig out the characteristics of the implicitness of infringement, the expectation of implementation, and the concreteness of aspects. On this basis, the regulatory path of the infringement of GIs of agricultural products is put forward, such as introducing procuratorial public interest litigation, multi-agents cooperating to implement all-round supervision, and reasonably determining the amount of damages.

## 1. Introduction

With the improvement in people’s quality of life and the transformation of consumption concepts in all countries, healthy consumption is becoming the highlight of the recovery of the consumer market. Consumers are paying more attention to food quality and safety and are willing to pay for the transparency and traceability of food [1]. The place of origin is used as a quality signal, which makes products with professional quality stand out among a large number of ordinary products [2]. With the support of policymakers and the market, producers and other participants establish the connection between origin and food quality to meet consumers’ demand for healthy food [3] and convey information to consumers through geographical indication labels (GIL), which reflect both origin and quality assurance [4]. Although food with geographical indications is sold at a higher price in the market, consumers are willing to pay extra for the reputation and quality of food [5,6], so geographical indications (GIs) provide producers with opportunities to improve their competitiveness [3].

Geographical indication is a controversial issue in trade negotiations [7]. For the protection of geographical indications (GIs), countries such as the European Union’s member states are in favor of a special system mode of protecting products according to geographical origin, while other countries represented by the United States support the trademark law [8]. Although a variety of protection modes of geographical indications have emerged in the world, which provide rich reference and experience, it is still a new topic for developing countries [9]. GIs were imported into China, and the protection system developed relatively late, but China has actively explored in practice and creatively established a unique protection system for geographical indications on the basis of the above two models, namely, “two models and three systems” [10]. Two models refer to trademark law and special law, while three systems refers to collective trademark and certification trademark system, protection system of geographical indications of agricultural products, and protection system of geographical indications products. In addition, “two models and three systems” are supplemented by unfair competition law. The legal protection of geographical indications of agricultural products in China is still in the stage of continuous exploration. In the process of the exploration of protection methods, there are not only protection modes led by Trademark Law of People’s Republic of China (PRC), taking Anti-Unfair Competition Law of People’s Republic of China (PRC) as the base, but also systematic protection modes that integrate all existing protections [11]. At the legal level, on the one hand, the existing Measures for the Administration of Geographical Indications of Agricultural Products and the Regulations on the Protection of Geographical Indications can solve the rigid conflict in the process of protecting geographical indications of agricultural products to a certain extent [12]. On the other hand, enacting a special law with a higher legal level, such as the Law on the Protection of Geographical Indications, can also effectively deal with the problems of low legal level, chaotic management, and frequent conflicts in the current process of protecting geographical indications of agricultural products, thus making the protection work systematic. The geographical indications of agricultural products are different from other traditional intellectual property rights; therefore, the problems they face are more complicated when identifying the attributes of infringement [13].

In China, as a traditional agricultural country, geographical indications of agricultural products have always been a strong point in its intellectual property field, which is of strategic significance to economic and social development. According to the report of the 20th CPC National Congress, the Communist Party of China (CPC) has made a major decision-making arrangement of “speeding up the construction of an agricultural power and consolidating the foundation of food security in an all-round way”. As a high-level platform for China’s agricultural products to take the brand path, geographical indications of agricultural products have great potential space for helping China to take the path of agricultural power with distinctive characteristics. With the vision of developing local characteristic brands to promote economic and social development, the number of applications for geographical indications of agricultural products in China has also risen sharply. According to the recent statistics of GIs released by the General Assembly of China’s State Intellectual Property Office, by the end of June 2022, China had approved a total of 2493 products for GIs protection and 6927 registered GIs as collective trademarks and certification trademarks (The data come from geographical indication network: http://www.cpgi.org.cn/?c=index&a=detail&cataid=1&id=4012, accessed on 2 October 2022). However, with the continuous expansion of the market of geographical indications agricultural products and the growing prosperity of the industry, a large number of infringements of counterfeiting and misusing of geographical indications of agricultural products have also appeared. As of 31 July 2022, the number of intellectual property and competition disputes with “GIs” as the full-text keyword in “PKULAW”, which is a one-stop search platform for legal information, exceeded 2000. There are countless cases of infringement of well-known geographical indications for agricultural products such as Anji White Tea, West Lake Longjing Tea, and Shaoxing Yellow Wine. The decline in sales volume and credibility, coupled with the high cost of safeguarding rights, has caused heavy losses to enterprises and farmers, disrupted the order of fair markets, and caused great damage to the brand reputation of geographical indications agricultural products. counterfeit products also have the possibility of not meeting the food safety standards, which can easily to bring about food safety risks. Because of the particularity of the infringement of geographical indications of agricultural products, the cost of safeguarding the rights of the holders is extremely high. Compared with the common trademark infringement cases, the input cost of each link of safeguarding rights is higher, and the actual operation is more complicated. Understanding how to better regulate the infringement of geographical indications of agricultural products has become an inevitable requirement for maintaining the brand effect and taking the path of strengthening the country with characteristic brands. For the infringement of the geographical indications of agricultural products itself, the immateriality of the infringing object means that the infringement is different from the traditional infringement [14]. First of all, the immateriality of the infringing object of geographical indications of agricultural products determines that the obligee is in an awkward position when defending the infringement independently; on the one hand, the premise of obtaining the right means making the object of rights public to a certain extent, but once the object of rights is made public, the obligee will lose material control, and anyone may copy and spread it. As a result, the low cost of infringement leads to frequent infringements [15]. Secondly, although the geographical indications of agricultural products are generally within the protection scope of the trademark law at this stage, in actual infringement cases, it is challenging to define the object of infringement. Specifically, if we delve into the significance of geographical indications of agricultural products, opinions vary on this issue. The interpretation space of many concepts about the infringement of geographical indications of agricultural products is obviously broader than that of tangible property in traditional infringement law [16]. Thirdly, the determination of the amount of damages is definitely an important issue in tort, but in judicial practice, the damage of the relevant intellectual property rights caused by the infringement of geographical indications of agricultural products is often impossible to verify and quantify; therefore, understanding how to calculate the amount of damages in the event becomes a tough problem. Fourthly, the impact of the infringement of geographical indications of agricultural products is often related to public interests. The exclusive right comes from legal systems such as trademark law, which aims to encourage the creation, dissemination, and use of knowledge and wisdom so as to promote policy judgments of social progress. Thus, disputes about public interests often have an impact on the judgment and bearing tort liability of geographical indications of agricultural products [17].

On this basis, this paper takes “the infringement of GIs of agricultural products” as the research object, systematically sorts out the judicial patterns of the disputes over the infringement of GIs of agricultural products in China, sums up the main points of litigation after sorting out the types, explores the judicial profile of the infringement and the overall outline in practice, and considers the dimensions of the infringement of GIs of agricultural products to dig out the characteristics of the infringement, so as to clarify the judicial deviation of the infringement of GIs of agricultural products. The following three questions are addressed: first, whether there are obvious laws in the judicial cases of infringement of GIs of agricultural products, such as the type of plaintiff, the respondent, and the application of law; second, what the controversial points are in the regulation of infringement of GIs of agricultural products; third, how to reverse the judicial deviation of the infringement of GIs of agricultural products.

The rest of the article is carried out according to the following structure. Section 2 puts forward the shortcomings of the existing research system on the infringement of GIs of agricultural products and the innovation of this article on the basis of summarizing the existing literature; Section 3 introduces the samples selected in this paper and the quasi-case research method adopted and deconstructs the samples on this basis; Section 4 sorts out the types and summarizes the main points of contention on the basis of sample deconstruction and considers the infringement of GIs of agricultural products in different dimensions; Section 5 puts forward effective measures to regulate the infringement of GIs of agricultural products; and Section 6 restates the research significance of the article while summarizing the conclusion and looks forward to the future research.

## 2. Literature Review

The European Union regards GIs as a tool in rural development strategies to promote rural development and ensure cultural and biological diversity [18,19,20]. From the perspective of global value, GIs promote economic development, increase profit income, enhance collective production power, improve the positioning of small-scale producers, and provide diversified development modes [21]. From the perspective of social welfare, GIs not only improve people’s profitability due to higher premiums over the cost of labeling but also help poor areas to maintain the number of farms and arable land, therefore retaining the rural population [20]. From a cultural development point of view, the industrial organization to which GIs belong is the main agent for constructing local cultural characteristics [22,23]. From the perspective of consumption, the emergence and development of GIs reduce the information asymmetry between consumers and operators and create conditions for sustainable consumption [24]. From the food security point of view, a large number of scholars have found, by building an empirical model, that the brand-building of GIs has a positive impact on the quality and safety of agricultural products, which is conducive to the green and high-quality development of agriculture [25]. From the perspective of sustainable development, GIs are strategic tools [26] to promote sustainable rural development in many respects, such as social, economic, and environmental factors [27], and to prevent the consequences of agricultural industrialization or ecological destruction [28].

With globalization, the risk of misappropriation of place names is increasing [29], and the infringements of the illegal use of GIs are increasing [30]. GIs are mostly regarded as a kind of public resources [31,32], and hitchhiking is common in the field of GIs. The price premium of products with GIs leads to some opportunistic behaviors, such as mixing Indian fragrant rice GI products with other varieties of rice [33]. At the same time, there is always a dispute between GIs and generic names of products in practice [34]. The illegal use of the name of origin not only damages the reputation of GI products but also deprives the real right holder of the return on investment in the development of goods and the reputation of the product market and greatly misleads consumers towards buying counterfeit GI products [35]. The degree to which the names of GIs are counterfeited by competing products depends on the intensity of GI protection [36]. It is an important and complex task for relevant agencies to monitor the use of GIs and regulate the infringement of GIs [30]. The establishment and maintenance of GI product quality standards, rules for the use of GIs, and the supervision and regulation of GI infringement will help to safeguard the reputation of GIs and effectively eliminate illegal acts. Meanwhile, it will promote the empowerment of GIs, enhance the trust and social cohesion of stakeholders [37], and ease the internal and external pressure on GI systems [38].

In recent years, more attention has been paid to the infringement of GIs worldwide [39]. So far, scholars mostly examine the nature of infringement of geographical indications of agricultural products from the perspective of GI trademark and GI protection system, such as the right boundary of GI certification trademark, infringement judgment standard, and proper use. Focusing on GI trademarks, scholars have investigated the essence of the conflict between protecting GIs and strengthening special property rights of GIs under the trademark law system by analyzing the legal nature and scope of protection of GIs and general trademarks, and they have also explored the relevant dilemmas from the current situation of judicial application of trademark infringement identification standards of GIs [40]. Focusing on the study of the protection system of GIs, scholars have mainly discussed the emergence and evolution of the historical and international demand for the protection of GIs [41] and sorted out and summarized the problems existing in the system and the reasons for the challenges in the protection of GIs in a comprehensive way [42]. In addition, they have examined the multi-center governance method of GIs from the perspective of international comparison [43] and revealed the internal relationship between various stakeholders of GIs and the protection system of GIs through case analysis [44,45,46], so as to seek countermeasures and suggestions for optimizing the legal protection model of geographical indications of agricultural products, improving the effect of the system and perfecting practical implementation [28].

In the existing research on the protection of GIs, scholars have tended to ignore the analysis and re-understanding of the nature of the infringement of geographical indications of agricultural products. The protection of GIs and the regulation of infringement must start from the essential attributes of infringement and combine the important and difficult points in the logic of judicial practice in order to find a more accurate regulation. In this paper, the quasi-case-study method is used to sort out the disputes and clarify the judicial practice logic of infringement of geographical indications of agricultural products in China. Based on further uncovering and sorting out the types, the essential characteristics of the infringement are analyzed. Then, the paper reflects on the controversial issues in China’s judicial practice regulation and explores the effective regulation strategies for the infringement of geographical indications of agricultural products. Existing literature on the GIs of agricultural products is shown in Table 1.

## 3. Materials and Methods

Quasi-case research is a legal argumentation method based on case analogy. By integrating case facts, controversial focus, legal application, and other case elements, the similarity judgment of cases based on the legal argumentation model is realized. Among them, “quasi-case” refers to a case that is similar to the pending case in terms of basic facts, controversial focus, application of law, etc., and has come into effect through court judgment. Based on the quasi-case research method and typological thinking, this paper builds a bridge between the abstract concept of infringement of geographical indications of agricultural products and judicial practice, and based on induction and deduction, it explores the judicial profile of the research object in this manuscript on the basis of sample deconstruction. It should be noted that in common-law countries, the rules of precedent and the rules of enactment are parallel to each other in form, and the former has a far greater influence on events than the latter in essence. In China, although quasi-cases do not have universal binding force in the abstract sense such as legal or judicial interpretations, it is difficult to clarify the meaning of similar cases with the help of a single precedent, and its binding force goes beyond the case itself in a certain sense; consequently, it becomes an important judgment basis for pending cases in the absence of a clear basis such as laws and judicial interpretations. On this basis, this paper uses “PKULAW” as a retrieval tool to make statistics on the civil cases of infringement of geographical indications of agricultural products in China from 1 January 2014 to 31 July 2022 (The Trademark Law of People’s Republic of China (PRC) was amended in 2014, so the cases were selected from 2014 onwards). In order to fully show the patterns of infringement of geographical indications of agricultural products in China, this paper sets different retrieval conditions for two searches on the selection of analysis samples. For the first time, the “geographical indications of agricultural products” were used as the key words in the “full content” section, and 104 samples of judgments were obtained. In the second search, the keywords “GIs”, “agricultural products”, and “trademark disputes” were searched in the section of “all contents”, and 281 samples of judgment were obtained. There are many unrelated samples in the retrieval cases or disputes about the application of geographical indications of agricultural products, or references to geographical indications of agricultural products when explaining the basic facts of the case, but none of them refers to the infringement of geographical indications of agricultural products discussed in this paper. A total of 245 valid samples were obtained after screening (A total of 385 refereeing documents were obtained from the two searches, and 301 samples were obtained after eliminating duplicate documents. Among them, there are still 8 disputes involving geographical indications or trademark applications for geographical indications, 20 disputes involving contracts for the sale of agricultural products with geographical indications, 2 disputes involving the determination of common names between ordinary trademarks and geographical indications of agricultural products, and 26 disputes involving geographical indications of agricultural products only when describing the basic facts of the case, which are not related to the infringement disputes of geographical indications of agricultural products). The analysis path is shown in Figure 1.

The number of cases from 2014–2022 was 10, 15, 13, 22, 5, 39, 56, 74, and 11 for each respective year. The numbers increased in a wave-like manner, especially in 2015 and 2020, which showed that the amendments to the Trademark Law of People’s Republic of China (PRC) in 2014 and 2019 stimulated the parties’ enthusiasm for prosecution to a certain extent. The number of prosecutions in 2019, 2020, and 2021 increased significantly, indicating that the infringement of geographical indications of agricultural products has attracted the attention of the society, and the decrease in the number of cases in 2022 due to unfinished cases, untimely uploading, downward economic pressure, and the impact of the pandemic is also reasonable. It can be predicted that with the general development and in-depth promotion of special protection actions for GIs products in various regions, the number of cases of infringement disputes over geographical indications of agricultural products will increase day by day. The specific cases involved in the article are listed in Table 2 (the original data of 245 cases are submitted in the Appendix A).

### 3.1. The Type of Plaintiff Presents Double Simplification

After sorting out 204 samples of the first instance, it was found that 198 of them, accounting for about ninety-seven percent, were lawsuits filed by industry associations that are the registrants of GIs. There were two lawsuits filed by the industrial station of the county agriculture bureau, two filed by the provincial agricultural technology extension center sued, one filed by the Office of the Industry Management Committee (the Industry Head Office), and one filed by the limited liability company as the holder of the right to use the trademark (Table 3). The geographical indications of agricultural products have three kinds of subjects of rights: the ownership holder, the right holder, and the prohibition rights holder. The geographical indications of agricultural products are bred in the blend and evolution of the long-term ecological environment and human history. Even if they are registered as trademarks, they should still be shared by the people in the agricultural areas. Therefore, the people in the agricultural areas should be the owners of the geographical indications of agricultural products. In addition, the holder of the right of use should be the producer and operator of the agricultural products, while the holder of the right to prohibition should be the registrant of the geographical indications of agricultural products. As for the disputes caused by the infringement of geographical indications of agricultural products, in judicial practice, it is almost always the holder of the right of prohibition, i.e., the trademark registrant, that files a lawsuit. In addition, a lawsuit filed by the industry association as a registrant occupies an absolute position in all trademark registrant’s lawsuit cases, and the subject structure of the lawsuit as a whole presents double simplification.

### 3.2. From the Perspective of the Sued Subject, the Online Infringement Case Network Platform Providers Are Mostly Co-Defendants

Judging from the distribution of the types of defendants, individual businesses are the main tortfeasors, of which the cases of individual or joint defendants account for about 73.5%, which reflects the characteristics of easy implementation and low cost of GI infringement. In the online infringement cases of geographical indications of agricultural products, network platform providers such as Shanghai Dreamfinder Information Technology Co., Ltd. (Shanghai, China) and Hangzhou Alibaba Co., Ltd. (Shanghai, China) are the main co-defendants (Table 4). The plaintiffs list the platform provider as a co-defendant, but in the presentation of the litigation results, the platform providers were not required to assume any liabilities in the 18 cases. Actually, the platform providers were more used as evidence material to provide the operation of the shops involved and the links to the commodities involved. For example, in the first instance of a trademark infringement dispute among Hangzhou West Lake District Longjing Tea Industry Association, Songyang County Hongming Tea Company and Hangzhou Alibaba Advertising Co., Ltd., the court held in the reasoning section that “Alibaba Company, as the operator of the online trading platform, does not participate in the specific sales of goods, and cannot conduct substantive examination on a large number of goods except for qualification examination of merchants”. “After receiving the lawsuit materials, Alibaba Company took relevant measures. The alleged infringing products have also been taken off the shelves, and the West Lake Longjing Association has not raised any objection. It should be determined that Alibaba Company has prevented the expansion of the scope of infringement in a timely manner and has done its reasonable duty of care and supervision” (See Zhejiang Lishui Intermediate People’s Court (2018) Zhe 11 Min Chu No. 84 Civil Judgment.).

### 3.3. From the Perspective of Infringement Types, Marginal Infringement Is the Basic Form

The types of infringement are mainly divided into two categories: one is the unauthorized use or forgery of GI names and special signs; the other is the use of the names of GI products that do not meet the requirements of the GI product standards and management specifications. The most important difference between the two types of infringement is whether the agricultural products are from the GI’s area. Of the 204 cases of first instance, 201 cases were found to constitute infringement by the court, of which unauthorized use of forgeries was the most important type of infringement, with 189 cases accounting for 94.03% (Table 5). The specific implementation methods of such infringement mainly include similar a product name, a similar trademark, and the same origin. For example, “the product name has a large number of words, without distinction between primary and secondary, and the word ‘origin’ is used between ‘Xinyang’ and ‘maojian’,” (See Beijing Xicheng People’s Court (2021) Jing 0102 Min Chu No. 16099 Civil Judgment) and “the origin of raw grain is highlighted as the high-quality rice production base in Panjin, Liaoning province” (See Tianjin No. 1 Intermediate People’s Court (2017) Jin 01 Min Chu No. 405 Civil Judgment). The actual infringement is a collection of the various above-mentioned types. The edge of infringement structure in various respects has some connection with GIs agricultural products, which causes consumers to misidentify or confuse the products, thus realizing the purpose of profit-making.

### 3.4. From the Perspective of the Application of Law, the General Trademark Provisions Occupy a Dominant Position

Through the statistical court’s trial of 201 cases of infringement samples cited in the first instance, the legal provisions related to infringement identification are sorted according to the number, which can preliminarily show the legal application of infringement identification of geographical indications of agricultural products. Among them, the most cited provision is Article 57 of Trademark Law (Table 6). Article 57 of Trademark Law is a general provision of common trademarks, which stipulates the types of behaviors that infringe on the exclusive right of registered trademarks. It plays a role in identifying infringement behaviors in current GI infringement disputes, but it is still weakly related to the identification standards of infringement of geographical indications of agricultural products in essence. From the legal point of view, the adjustment object of this provision does not cover GIs. The differences in origin and quality between products identified by common trademarks and by GIs will lead to different types and results of infringement; therefore, the standards for identifying infringement are different. The judicial authorities adopt this provision to identify the infringement of geographical indications of agricultural products. First, it is beneficial to the integrity of the trademark law system when it is proved that GI trademarks are trademarks in essence; therefore, the infringement of identification standard is equivalent to that of general trademark standards. Second, there is no legal norm to specifically regulate the infringement of GIs at present. Limited by the lack of special laws and regulations, judicial organs can only invoke related laws and regulations, but it is difficult to provide specific special rules and guidelines for the handling of specific disputes.

### 3.5. Judging from the Compensation Standard, Abstract Factors Are the Main Determination Standard

The factors considered in determining the amount of compensation in the case of infringement disputes over geographical indications of agricultural products include rule factors and individual factors. The second paragraph of Article 16 of the Interpretation on Several Issues Concerning the Applicable Law in the Trial of Trademark Civil Dispute Cases (“When determining the amount of compensation, the people’s court shall take into account the nature, duration and consequences of the infringement, the reputation of the trademark, the amount of trademark license fee, the type, time and scope of trademark license and the reasonable expenses to stop the infringement.”) specifies the rule factors that need to be considered. Except that “reasonable expenses to stop the infringement” can be directly used as a specific standard to determine the amount of compensation, the applicable standards of other factors are not detailed. They are quite abstract in the application of judicial practice and require a high degree of judgment and life experience from the judge. When applying the abstract rule factors, the court is usually vague and only makes a general list, which is generally summarized by “comprehensively considering the factors such as the reputation of the trademark in dispute, the circumstances of the infringement, etc.” At the same time, the number of cases considering individual factors is small (Table 7). For example, the judge takes the nature of the infringer and the attitude of the infringer to correct errors into consideration; e.g., “Especially considering that Feng Mohui’s business department is a seller of various varieties of chili peppers, it is determined as appropriate” (See Hunan Changsha Yuelu People’s Court (2022) Xiang 0104 Min Chu No. 5266 Civil Judgment), and “The degree of fault and attitude of the infringer to correct errors are determined as appropriate.” (See Guangdong Guangzhou Tianhe People’s Court (2020) Yue 0106 Min Chu No. 12496 Civil Judgment) However, in essence, the amount of compensation is also vaguely determined to some extent.

## 4. Results

### 4.1. Type Arrangement: The Dispute Point of the Regulation of the Infringement of Geographical Indications of Agricultural Products

The infringement of geographical indications of agricultural products is obviously different from the general trademark infringement because of the special nature of the object of damage. In particular, the characteristics of the former, such as implicitness, abstraction of applicable standards, and difficulty in quantifying damages, have caused new difficulties in the regulation process and brought new challenges to the identification of infringement, the use of place names, and the identification of infringement liability in disputes over geographical indications of agricultural products.

#### 4.1.1. Disputes over the Cognizance of Infringement: It Is Difficult to Define the Validity of Evidence and the Standard of Geographical Indications of Agricultural Products

The difficulty of proof of infringement leads to disputes over the identification of infringement, which are shown in Table 8. At present, the way of obtaining evidence for an infringement of geographical indications of agricultural products is that the obligee buys counterfeit products under the notarization of a notary, and the relevant departments inspect the product marks and quality and issue notarial certificates and opinions. However, this method can easily fall into the predicament of phishing law enforcement, which becomes the defense reason of the defendant. For example, the appellant in the second instance of trademark infringement dispute between Longjing Tea Industry Association and Xu Jianfu in Xihu District, Hangzhou (See Guangzhou Intellectual Property Court (2015) Guangdong Zhi Fa Shang Min Zhong Zi No. 202 Civil Judgment), which is also the defendant in the first instance, said that “the evidence obtained by Longjing Tea Association by illegal means is not legal and should not be used as the basis for ascertaining the facts. Taking advantage of the characteristics of the bulk tea retail industry, it adopts fraud, seduction, fishing for evidence and sets up a trap for evidence “. The premise of the standard of misidentification is that the GIs of agricultural products enjoy high popularity, which is enough to make people associate counterfeit GIs with genuine GIs of agricultural products. Therefore, the rights holder of GIs further proves that the defendant’s behavior is enough to make people be mistaken about the origin and quality, thus constituting infringement. The proof of reputation is abstract. In the civil second-instance case of trademark infringement dispute between Li Minfu Rice Industry Co., Ltd. in Gushi County and Xiantao Rice Industry Association as well as Dashikahai Department Store in Panyu District, Guangzhou (See Guangzhou Intellectual Property Court (2020) Guangdong 73 Min Zhong No. 4846 Civil Judgment), the appellant (the defendant in the original trial) claimed that the evidence given by Xiantao Rice Industry Association about the popularity of the trademark involved in the case was only an honor in 2012 and 2015. This evidence cannot directly prove that “Xiantao fragrant rice” products still maintain a high reputation in the market, and the relevant public is more cautious in choosing and judging rice products closely related to health and will not identify the source and quality of goods just by the “peach” pattern.

The vague standard of GI products leads to disputes over the identification of infringement. Standards are the basis of confirming the rights of GIs. They occupy the core position in the complete logical chain of “standards of GIs products → specific quality characteristics of GIs products → reputation of GIs → fundamental interests of GIs owners”. They are the tickets for producers to enter specific markets and the “signal” to convey product quality assurance to consumers [47]. Therefore, for agricultural products produced with GIs, the use of GIs product names that do not meet the requirements of GI product standards still constitutes GIs infringement. In China, there are many problems in the standardization system of GI products, such as lagging behind in the formulation of product standards, the imperfect standard system, and the lack of standards or the presence of unclear standards of the GIs of agricultural products most of the time, which leads to disputes over the identification of infringement. First, the standard formulation is lagging behind. After geographical indications of agricultural products are officially approved by the State Intellectual Property Office, the national or local standards are formulated on the basis of the draft of enterprise standards, group standards, or local standards attached to the application stage. However, according to the statistics of the State Intellectual Property Office and the national standard information public service platform, as of the end of June 2022, China had approved a total of 2493 GIs products, while there were 2005 standards related to GIs products, which indicates that some GIs products standards had not been officially issued. There is a certain time lag between the formulation of the official standard of GIs products and the protection of GIs products [47]. The time period before the official standard is introduced is a weak area in the regulation of the infringement of geographical indications of agricultural products that do not meet the standard. Second, the standard system is not perfect. Geographical indications of agricultural products have to go through raw material selection, production and processing, sales management, quality control, and other links from production to sales. However, at present, most GIs’ agricultural products have not established a standard system covering all links. In the standard system of GI agricultural products, in particular, there is a lack of standards for key links in the industrial chain, such as seed and seedling standards, planting technology regulations, breeding technology regulations, production and processing technology regulations, product environmental requirements, hygiene standards, and product quality traceability, which determine the quality of GI products as a whole [48]. The backwardness of standardization construction of GIs products leads to more infringements “taking advantage of hoopholes”. A typical case is the recently concluded civil first-instance case of infringement of trademark rights between Pepper Industry Association of Zhangshu Town, Xiangyin County and Yichuan Vegetable Store in Yuelu District, Changsha City (See Hunan Changsha Yuelu People’s Court (2022) Xiang 0104 Min Chu No. 5266 Civil Judgment). Defendant II denied the existence of infringements, arguing that “the trademark involved in the case is a ‘certification trademark’, and the ‘conditions of use’ formulated by the respondent are unclear, not explicit, unenforceable and actually not monitored”. It is believed that in this case, if even the so-called authentic GI commodities cannot be guaranteed to come from a specific place of origin, and there are no definite quality characteristics, the respondent cannot monopolize a local commodity name in the public domain. The court of first instance did not respond to this situation in the judgment and still judged that defendant II constituted infringement by using the trademark, avoiding the substantive issue of GIs as proven trademarks.

#### 4.1.2. Dispute over the Use of Place Names: The Standard of Legal Use of Common Similar Products in the Same Producing Area Is Vague

There may be infringement of GIs in the labeling of the origin of similar agricultural products; the specific disputes are shown in Table 9. Unlike the restrictions on place names in ordinary trademark registration, the Trademark Law, Measures for the Administration of Geographical Indications of Agricultural Products, and Regulations on the Protection of Geographical Indications of Products protect the use of place names in geographical indications of agricultural products. Then, in reality, the legitimate use of place names of similar agricultural products is possible. In the case that GIs already exist in agricultural products of the same origin, the law lacks clear provisions on the use of place names of similar agricultural products, and there are different practices in judicial practice. In the case of trademark infringement dispute between Wuchang City Rice Association, Li Guitong, and Beijing Jinli Xingsheng Cereals and Oils Trading Co., Ltd. (Beijing, China) (See Beijing Intellectual Property Court (2015) Jing Zhi Minzhong No. 1180 Civil Judgment), the defendant prominently marked the word “Wuchang” at the top middle of the front of the rice package and argued that “Wuchang” was a place name and therefore that it was a proper act to mark the place of origin on the commodity package. Although the trademark involved was not similar to the GI trademark of Wuchang Rice, the court held that the way of marking the place of origin was improper, which caused consumers to mistake it, and still determined that the defendant was responsible for infringement. In the first-instance case of trademark infringement dispute between Apple Association of Aksu Region and Beisong Fruit Firm of Xining City (See Qinghai Xining Intermediate People’s Court (2020) Qing 01 Zhi Minchu No. 38 Civil Judgment), the court held that although logos such as “Aksu Apple” and “Aksu” used by the defendant on the commodities involved were not exactly the same as the trademarks involved, the words of the trademarks and the prominent place they are marked on the commodities would make the relevant public think that the apples involved were apples originating in Aksu region. Therefore, if the goods involved did not originate in Aksu, the Apple Association of Aksu has the right to prohibit the defendant from using the words “Aksu Apple” or “Aksu”, which describe the origin of the goods, and investigate the defendant’s infringement liability accordingly. On the contrary, if the goods involved originated in Aksu, the defendant’s behavior is legal. The appellate court of the trademark-infringement dispute between Apple Association in Aksu and Xingmin Vegetable and Fruit Firm in Xining Chengbei (See Qinghai Provincial Higher People’s Court (2020) Qing Zhi Minzhong No. 20 Civil Judgment) also recognized the trademark applied for registration by Apple Association in Aksu, which was a GI to prove the registration of the trademark and cannot deprive the firm of the right to properly use the place name of the certification trademark. The above-mentioned Aksu Apple case and Wuchang Rice case have similarities, but the verdict is opposite. The similarities involved in the cases include the use of trademarks different from the certification trademarks, the places of origin being GIs, and the places of origin being marked in an obvious way. The courts all think that this will cause consumers to misunderstand the origin and quality of commodities. However, in the final judgment, Wuchang Rice Court ruled that the defendant’s behavior constituted infringement, while Aksu Apple Court held that the defendant’s behavior was justified.

#### 4.1.3. Disputes over Infringement Liability of Geographical Indications of Agricultural Products: The Amount of Damages Is Abnormally Light

Infringement compensation is the most important way to assume responsibility. According to Article 63 of China’s current Trademark Law, the order of compensation for damages of infringement of trademark rights is as follows: the actual losses suffered by the obligee due to infringement → the interests gained by the infringer due to infringement → the multiple of trademark license fee → the compensation below CNY 5 million [40]. In most cases, it is difficult for the plaintiff to prove the amount of loss suffered by the defendant due to infringement or the amount of benefit gained by the defendant due to infringement, and the license fee of the disputed trademark cannot be referred to. Local courts usually use the fourth compensation order and adopt the statutory compensation method when making judgments on damages. Because of the abstractness of the standard for determining the amount of compensation, the court is often more cautious in awarding compensation, which leads to an abnormally low amount of damages. Specific disputes are shown in Table 10.

With 23 cases of infringement disputes of Longjing in West Lake taken as an example (Table 11), in the case of infringement of trademark rights between Longjing Tea Industry Association in West Lake District, Hangzhou, and Jianming Tea Shop in Yangpu District, Shanghai (See Shanghai Yangpu District People’s Court (2016) Hu 0110 Min Chu No. 3624. Civil Judgment), the plaintiff asked the defendant to eliminate the influence and compensate the plaintiff for the economic loss of RMB 50,000, including the reasonable expenses to stop the infringement. In the end, the court rejected the petition to eliminate the impact and ordered the defendant to compensate the plaintiff for the economic loss of RMB 18,000, including the reasonable expenses of RMB 1700 paid by the plaintiff to stop the infringement, and the amount of support damages was 36% of the claimed amount. In the reasoning part of the court’s judgment, it was explained that the relevant materials submitted by the plaintiff were not enough to prove that the defendant’s infringement had caused adverse effects on its trademark goodwill, so it did not support its claim of eliminating the effects. Although the plaintiff claimed compensation for the losses, it failed to prove the actual losses suffered by the defendant’s infringement or the profits gained by the defendant’s infringement. Therefore, the amount of compensation was determined as appropriate by comprehensively considering the popularity of the trademark involved; the defendant’s business scale; the nature, circumstances, and consequences of the infringement; etc. There are more cases in which the claimed amount of infringement compensation is quite different from the judgment amount and the judgment amount is obviously lower than that expected by the plaintiff. For example, the judgment ratio of the first-instance case of trademark infringement dispute between Longjing Tea Industry Association in Xihu District of Hangzhou and Hongming Tea Company in Songyang County and Hangzhou Alibaba Advertising Co., Ltd. (Hangzhou, China) (See Zhejiang Province Lishui Intermediate People’s Court (2018) Zhe 11 Min Chu No. 84 of People’s Republic of China. Civil Judgment) is 12.5%, and the amount of compensation of the trademark infringement dispute case between Longjing Tea Industry Association in Xihu District of Hangzhou and Fuxin Grocery Wholesale and Retail Store in Yinzhou District of Tieling City (See Liaoning Province Tieling Intermediate People’s Court (2017) Liao 12 Min Chu No. 94 Civil Judgment) is only RMB 3000, which accounts for only 3%. The degree of infringement liability, especially the liability for damages, is seriously insufficient, which leads to the increase in the rate of the second instance and the waste of judicial resources, and it is difficult to deter other GIs infringers.

### 4.2. Dimension Consideration: Judicial Deviation of Infringement of Geographical Indications of Agricultural Products

Through sample deconstruction, the judicial status of infringement disputes of GIs is preliminarily revealed. Next, the infringement itself is stripped from judicial cases, and the characteristics of infringement are further summarized and sorted out in combination with the particularity of geographical indications of agricultural products.

#### 4.2.1. Deviation between the Concealment of Tort and the Explicitness of Object of Tort

The object of infringement of geographical indications of agricultural products is a special type of intellectual property rights, and its infringement is concealed, which is different from infringement of personal rights and general intellectual property rights. Compared with the infringement of personal rights, the infringement of geographical indications of agricultural products is widely distributed and scattered, so it is not easy to monitor; in addition, the resulting harm is not significant. Taking Korla fragrant pear as an example, among the 29 samples of first-instance infringement cases of the Korla fragrant pear, the infringements are distributed in Henan, Beijing, Shandong, Jiangsu, Sichuan, Zhejiang, Heilongjiang, Shanghai, and Guangdong provinces (Table 12). This shows that infringements are widely distributed and scattered, and industrial associations therefore have limited ability to organize and supervise, making it difficult to detect the existence of infringements. When the infringement of agricultural products with counterfeit GIs is carried out, consumers will buy counterfeit geographical indications of agricultural products and question the quality of genuine geographical indications of agricultural products after consumption, which will reduce the purchase of geographical indications of agricultural products and reduce the sales of genuine products. However, the decrease in commodity sales is often the result of multiple factors, and the damage caused by infringement is hidden and difficult to detect. The infringement of geographical indications of agricultural products is also different from the infringement of general intellectual property rights, which usually involves the unauthorized dissemination and use of works without the permission of the rights holder, and behavior is therefore irrefutable once discovered. However, in some cases, it is difficult to judge whether an act constitutes infringement of geographical indications of agricultural products. For example, for “Rice of Wuchang city “ and “Wuchang Rice”, the Wuchang city marked by the former refers to the place of origin, while the latter is the formal geographical indication of agricultural products. The implementation of GI infringement is hidden. Producers or operators make consumers mistakenly think that commodities are geographical indications of agricultural products through misleading language and signs playing the “edge ball”. Additionally, “marking the place of origin on the packaging of commodities is a legitimate act” is often the reason for defense.

The infringement of geographical indications of agricultural products encompasses a wide range of victimized groups and damages. In the sample of the first trial, the prosecution subjects include the industrial association as the trademark registrant of the GI certificate, the industrial station of the county agricultural bureau, the provincial agricultural technology extension center, and the office of the industrial management committee. These subjects are the direct objects of the infringement of geographical indications of agricultural products. At the same time, the infringement of geographical indications of agricultural products indirectly reduces the income of producers of geographical indications of agricultural products, damages the legitimate rights and interests of consumers who buy counterfeit geographical indications of agricultural products, and destroys the market order of fair trade. In particular, infringers evade market supervision and practice counterfeiting. It is quite possible that agricultural products with counterfeit GIs do not meet the food safety standards and therefore cannot meet the people’s demand for safe and nutritious food, which also creates a great crisis for social food security. In addition, the geographical indications of foreign agricultural products registered in China cannot escape the fate of being counterfeited.

#### 4.2.2. Deviation between Anticipation of Infringement and Hysteresis of Geographical Indications of Agricultural Products

The time at which the infringement of geographical indications of agricultural products occurs has its particularity, which is manifested in the fact that the infringement can be ahead of the registration time of the GI and the listing time of GI agricultural products.

The infringement of geographical indications of agricultural products can be ahead of the registration time of GIs. Formed in long-term development, geographical signs are closely related to the natural environment or human history; as a result, they are historic, scarce, and non-renewable. The trademark registration system cannot “create” a GI that does not exist in history. Even without registration, we cannot deny its existing status as an objective fact. This shows that even unregistered geographical indicators of agricultural products may be infringed. In June 2008, the Trademark Review and Adjudication Board of the State Administration for Industry and Commerce protected the unregistered geographical indicators of agricultural products in the trademark dispute case of Xianglian. In this case, “Xianglian” refers specifically to lotus seeds produced in Hunan Province. Although it had not been registered as a GI, it had obvious GI attributes and met the GI conditions stipulated in the second paragraph of Article 16 of the trademark law, and accordingly can be recognized as a geographical indicator of lotus seed products. However, the company was in Jianning, Fujian Province. Registering “Xianglian” as a trademark was suspected of misleading the public, which constituted an infringement of geographical indicators of agricultural products. This was the first time that the Trademark Review and Adjudication Board had identified GIs in a trademark dispute case, and this case expanded the new idea of GI protection. Since then, in the practice of trademark law, the censors and courts have also understood and applied Article 16 of trademark law in this sense and recognized this provision as the protection norm of unregistered GIs in the trademark law, that is, legally recognized the objectively existing social relations to protect unregistered GIs [25]. In the subsequent administrative dispute appeal case between the limited liability company Anhui Guorun Tea Industry and Qimen Black Tea Association of Qimen County (See Beijing Higher People’s Court (2017) Administrative Judgment No. 3288) concerning the request for invalidation of trademark rights, it was unreasonable for Qimen Black Tea Association of Qimen County to artificially change the reality of the objectively formed production area of “Qimen Black Tea” in history and register the trademark of GI in advance. In this regard, the court pointed out in the judgment that applicants for trademark registration of GIs should have more obligations of good faith when submitting the application documents for trademark registration than ordinary applicants for trademark registration of goods and services.

The infringement of geographical indications of agricultural products can be ahead of the listing time of geographical indications of agricultural products. As special food products of GIs, the production of geographical indications of agricultural products is more closely related to natural factors, which are often produced in a specific period of time. Counterfeit geographical indications of agricultural products have flooded the market before the real products are put on the market. For example, Korla fragrant pears are available on the market in large quantities only in mid-September, but among the 29 samples of first-instance infringement cases of Korla fragrant pear, 18 samples involved in counterfeiting products had sold pears in the market before mid-September. As a result, the products with various standards of counterfeit geographical indications of agricultural products seize the market, which affects consumers’ choice and trust of geographical indications of agricultural products, resulting in the phenomenon that “bad money drives out good money” (Figure 2).

#### 4.2.3. The Deviation between the Concreteness of Tort Behavior and the Abstraction of Tort Identification Standards

There are two views in academic and judicial circles on the standard of infringement of geographical indications of agricultural products: the standard of misidentification and the standard of confusion. The cases that adopt the misidentification of standard mainly focus on the consumers’ “misidentification” of the origin and quality. For example, the court found the infringement by “the relevant public will think that the rice involved is Panjin rice originating in a specific producing area” (See Tianjin No. 1 Intermediate People’s Court (2017) Jin 01 Min Chu No. 404 Civil Judgment). Cases that adopt the confusion standard are analogous to ordinary trademark infringement, and the judgment standard of infringement is that consumers “confuse” producers. For example, the court ruled that “the trademark involved in the case is similar to the plaintiff’s, and it is easy to confuse the relevant public about the product source” (See Shanghai Xuhui People’s Court (2021) Hu 0104 Min Chu No. 12355 Civil Judgment), which therefore constitutes an infringement. The application of the two standards has been controversial; so far, no unified instructions have been made in judicial practice, so the phenomenon of vague infringement standards has appeared in judicial practice. In the first-instance case of a trademark infringement dispute between Longjing Tea Industry Association in Xihu District of Hangzhou and Zhejiang Zhuoya Supply Chain Management Co., Ltd. (See Zhejiang Yueqing People’s Court (2020) Zhe 0382 Min Chu No. 5440 Civil Judgment), the court of first instance stated in the judgment that “in view of the popularity of the trademark involved, the above-mentioned behavior is likely to confuse or mistake the source of the commodity for the relevant public” and made the judgment in a conservative way, thus achieving the effect of reducing appeals. It is worth mentioning that the plaintiff used both the facts and the reasons to determine the infringement to the greatest possible extent, such as “its behavior is enough to cause the relevant public to misunderstand and confuse the ‘Xinyang maojian’ tea” (See Guangdong Zhongshan No. 1 People’s Court (2021) Yue 2071 Min Chu No. 11902 Civil Judgment).

The infringement of geographical indications of agricultural products should be judged by misidentification. First, according to Article 10, paragraph 7, of the trademark law, “misidentification” is not allowed to be used as a trademark, which is deceptive and makes it easy for the public to be mistaken about the quality or origin of the trademark. “Confusion” is expressed in the second paragraph of Article 57 of the trademark law as “without the permission of the trademark registrant, using a trademark similar to its registered trademark on the same commodity or using a trademark identical to or similar to its registered trademark on similar commodities is likely to cause confusion”. “Confusion” pays more attention to the attributes of the trademark itself, aiming at the source of the product, while the “misidentification” object is directly aimed at the quality, place of origin, etc., which focuses on the specific quality of the product itself and is more in line with the characteristics of GIs [49]. Secondly, from the subjective purpose of the infringer, whose direct intention is to hope that consumers will link agricultural products with the place of origin and that the agricultural products have a specific quality unique to the place of origin, so as to achieve the purpose of profit, not to hope that consumers will confuse producers. According to the principle of “unity of subject and object”, the standard of misidentification is more appropriate.

Regardless of whether the confusion standard or the misidentification standard is adopted, these standards are based on “people’s cognition”, relying on people’s subjective judgment, and reflecting the subjective judgment made by the judge in relation with the trademarks involved. It is the judge’s subjective speculation whether the common people will misunderstand or confuse the trademarks subjectively, which adds a double subjective component. In particular, the standard of confusion applied to the infringement of geographical indications of agricultural products is more subjective and abstract than the standard of misidentification to a greater extent. The manufacturer who confuses the object in the standard is specific, while the doer mistakenly recognizing the quality of agricultural products in the standard is abstract, and hence the judgment is more complicated and abstract.

#### 4.2.4. The Deviation between the Clarity of the Qualitative Standard of Tort and the Fuzziness of the Quantitative Standard

Compared with other intellectual property rights, the brand dependence of GIs is high, the subject of infringement is complex, the means are hidden, and the audience is widespread. It is difficult to collect the evidence of infringement damages and quantify the damages of GIs infringement because of multiple factors.

Judging from the loss of actual income, the actual prices of GI agricultural products, such as Dendrobium huoshanense, are very different from those of counterfeits. If the loss caused by the decrease in sales volume of genuine products is used to calculate the compensation amount, the amount is huge, which may be too severe for the infringer to compensate. However, if the amount of compensation is calculated by the actual income of the infringer from the sale, it is small, which will lose its practical significance for the obligee to invest a lot of costs to safeguard their rights. Therefore, it is difficult to balance the amount of compensation for the damages caused by infringement to fairly safeguard the legitimate interests of the obligee and the infringer. However, as a precondition, the actual loss of the plaintiff and the profit of the defendant due to infringement is still unclear. In judicial practice, the plaintiff cannot prove the economic loss suffered by the infringement or the profit gained by the defendant due to the infringement. There are no exceptions in the 245 samples. Judging from the damage to the brand value, the GI brand greatly enhances the added value of products because of its high quality, uniqueness, and scarcity, and the products are highly dependent on the reputation of the GI brand. The reputation damage caused by infringement will seriously hinder the construction of the GI brand and reduce the overall competitiveness of products. Moreover, the damage to actual and expected interest caused by the reputation damage of GIs in judicial practice is difficult to quantify and prove, making it difficult to determine the specific amount of damages. No case was found in the 45 samples that explicitly quantifies the reputation damage of geographical indications of agricultural products as a compensation amount. Judging from the infringer’s ability to fulfill the obligation of compensation in judicial practice, the sued subjects are mostly freelancers, such as individuals or individual industrial and commercial households. Their legal awareness is not strong, and their business model is not standardized, in which retail trading has not obtained a lot of money, and most of them can only make a living and do not have a higher ability to perform the compensation. Because there is no clear reference standard, it is difficult for the judge to determine the specific amount of damages based on the business scale and ability of the infringer to pay the compensation; therefore, its discretionary scope is wide.

## 5. Discussion

As mentioned above, it is difficult to effectively regulate the infringement of geographical indications of agricultural products. Apart from the particularity of the behavior itself, there are also some reasons such as the lack of legal norms and inadequate relief in reality. In view of the dispute of infringement regulation in reality, it is necessary not only to improve the protection norms and mechanisms of geographical indications of agricultural products on the macro level but also to optimize the industrial organization construction of geographical indications of agricultural products on the micro level. The regulation path is shown in Figure 3.

### 5.1. Division between Subjective and Objective: Integrating and Establishing the Protection System of Geographical Indications of Agricultural Products with Relevant Factors as the Leading Factor

The relevance of agricultural products’ areas of production is the core element of geographical indications of agricultural products; hence, it is the necessary condition for their confirmation. The differences in concepts and policy orientations between subjective relevance and objective relevance lead to the judicial deviation of the infringement of geographical indications of agricultural products. Only by grasping the key element of relevance can the problems of confirming the right of the geographical indications of agricultural products and controlling the judicial deviation of infringement be solved. Subjective relevance holds that the relevance of geographical indications of agricultural products is based on the fact that consumers associate a specific agricultural product with a specific place of origin, which exists in consumers’ cognition, while objective relevance holds that the environment of the place of origin of agricultural products creates the specific quality and characteristics of agricultural products, and this causal relationship is a specific fact. The differences between the trademark law and the Measures for the Administration of Geographical Indications of Agricultural Products in the definition of geographical indications of agricultural products lead to the judicial deviation in the forms and behaviors of the corresponding infringement, and the distortion and conflict in the subjective and objective correlation factors in the system of confirming the right to geographical indications of agricultural products and monitoring and protection also lead to the judicial deviation in the implementation time and identification standards of the corresponding infringement. From the perspective of subjective relevance, the protection of geographical indications of agricultural products by trademark law represents the protection of reputation, based on whether the reputation is damaged and whether consumers are misled or deceived, which means that it is a subjective and relative protection of private rights. The Measures for the Administration of Geographical Indications of Agricultural Products and the Provisions on the Protection of Geographical Indications Products or the introduction of new special laws are based on mandatory standards, which are directly implemented and supervised by administrative organs. Any products that do not meet the standards, that is, have no objective relevance, are not allowed to use geographical indications of agricultural products, regardless of whether consumers are misled or defrauded. This is an objective and absolute protection of public rights. The difference between the conceptual basis of trademark law protection and special protection leads to the different scope of protection and legal nature, which can not be replaced by each other, but each has its own division of labor and cooperates with the other.

### 5.2. The Protection of Public Right: The Introduction of Procuratorial Public Interest Litigation in the Protection of Geographical Indications of Agricultural Products

The “tragedy of the commons” phenomenon in the field of geographical indications of agricultural products restricts the development of GI industries. Practical difficulties such as the scattered legislation and low effectiveness of the protection of geographical indications of agricultural products, the multi-headed and inefficient management, and the particularity of infringement lead to the difficulty for the subject of rights to safeguard these rights. The introduction of the protection of geographical indications of agricultural products into procuratorial public interest litigation is one of the ways to solve this problem. This way of protecting geographical indications of agricultural products is flexible and can flexibly protect the interests of the rights holders in the last link. It is the last line of defense to regulate the infringement of geographical indications of agricultural products. By means of special legislation, procuratorial public interest litigation is introduced in the field of geographical indications of agricultural products, which clearly grants the people’s procuratorate the right to sue, gives full play to the legal supervision function of the procuratorate, and takes advantage of the procuratorate’s strong litigation ability to fully safeguard rights and deter infringers to achieve the effect of social demonstration. The conditions and scope of the public interest litigation of the procuratorate should be clarified. The litigation conditions should be that the infringement of geographical indications of agricultural products seriously damages the public interest and the relevant entitled litigants are idle or unable to prosecute or have not found the infringement. The scope of the lawsuit includes the unauthorized use or forgery of geographical indications of agricultural products, certification marks, and collective marks of geographical indications of agricultural products; the use of agricultural GI marks that do not meet the product standards and quality requirements of geographical indications of agricultural products; the unreasonable refusal by the registrant of geographical indications of agricultural products to use the GIs of eligible subjects; the unreasonable authorization by the registrant of geographical indications of agricultural products to other subjects to use the GIs; etc. In addition, the situation that the consumers’ genuine consumption rights of geographical indications of agricultural products are impaired should be improved; the length of protection of the rights and interests of GIs of agricultural products should be extended; and the market of geographical indications of agricultural products should be better connected with consumers [50]. Furthermore, the litigation procedure should be standardized. Under the framework of the general provisions of the Civil Procedure Law, some special provisions should be made in the litigation of geographical indications of inspection of agricultural products for the public interest. For example, in the court of jurisdiction, taking into account the regionality and particularity of geographical indications of agricultural products, it should be under the jurisdiction of the court where the geographical indications of agricultural products are located, so as to facilitate the litigation of the right subject and the examination of geographical indications of agricultural products, so as to improve the efficiency of litigation [51]. A civil compensation fund management system should be established; part of the civil compensation of the procuratorate’s civil public interest litigation should be transferred as compensation for the people whose interests are damaged by geographical indications of agricultural products, so as to re-stimulate the vitality of the GIs industry of agricultural products, and the rest should be deposited into the industrial management foundation to fulfill the future costs of rights protection [50].

The introduction of procuratorial public interest litigation into the protection of geographical indications of agricultural products also represents the objective need of the ineffective judicial protection of geographical indications of agricultural products in reality, and the inherent need for the procuratorate’s own innovative public-interest litigation mode. An aim of future development is to construct a new mode of public interest litigation and broaden the protection channels of geographical indications of agricultural products.

### 5.3. Public–Private Cooperation: The Multi-agent Cooperation to Realize the Full Coverage of Infringement Supervision of Geographical Indications of Agricultural Products

GI infringement is widely dispersed and highly concealed, so it is difficult to achieve ideal results by only relying on the subject of GI rights alone. In particular, at present, the weak functions and powers of industrial associations and the continuous upgrading of infringement and counterfeiting technologies have brought many challenges to the protection of GIs. Multi-party cooperation and group development should be an important breakthrough to solve the dilemma of the regulation of infringements.

The government is the core regulatory body for the protection of geographical indications of agricultural products, and it should play a role in guiding industry self-discipline. In addition to top-level design, government departments should also encourage stakeholders of agricultural products within GI industries in various fields of production, processing, and circulation to establish relevant industrial organizations, and fully authorize industrial associations to carry out work in the formulation of product standards, production organization, standardized management, brand application, trademark protection, and other aspects. Industry associations should cooperate with the government to implement the policies of GI protection through the method of government authorization or handing over supervision opinions to the government and reach a strong interactive new cooperative relationship with the government, so as to jointly build a vertical supervision system with a two-way linkage between the government and industry associations. Industry associations should also strengthen cooperation in the protection and development of geographical indications of agricultural products to build a horizontal mode of cooperation and supervision. The main offices of various industrial associations are distributed all over the country, and each has its own strong regulatory coverage. For the infringement of other geographical indications of agricultural products found within the regulatory scope of a certain GI organization, the organization takes the initiative to undertake the obligation of notification and helps “allies” to preserve the infringement evidence in advance without increasing its own burden. At the same time, all industrial associations should jointly upgrade their anti-counterfeiting technology, and go online and offline together in daily supervision. Meanwhile, they should join the relevant departments of the State Intellectual Property Office to supervise the online trading platform and enhance the defense strength of the whole industry of GIs. In terms of the publicity of GIs, we should jointly carry out enhancing the publicity of agricultural products’ identification of GIs, improve consumers’ ability to identify counterfeits, and therefore build a solid line of defense for consumers in regulating the infringement of GIs. The main elements within the industry of geographical indications of agricultural products must coordinate together to stimulate the endogenous power of industrial development. Each stakeholder in the industry of geographical indications of agricultural products, registrants and users, producers, operators, sellers, and others, should broaden their information-sharing channels and build an information-sharing platform in the industry to release and determine the legitimate users of agricultural products’ GIs to accelerate the information transparency of the whole industrial chain of geographical indications of agricultural products, so that the infringers of agricultural products GIs have nowhere to hide. The modes of cooperation are organic combination, integration of upper and lower levels, linkage between left and right, and internal coordination, so as to jointly realize the full coverage of the supervision over the infringement of geographical indications of agricultural products.

### 5.4. Pre-Prevention: Reasonable Confirmation of Compensation Amount for Infringement of Geographical Indications of Agricultural Products

If the reform of the management and protection of agricultural GIs from the perspective of legislation is achieved, it will require a long period of legislation and a coordination period between the new and old laws. The “tragedy of the commons” phenomenon of the current agricultural GIs cannot be controlled in a timely manner and cannot meet the urgent need for the regulation of the infringement of geographical indications of agricultural products. Therefore, reasonably aggravating the liability of the infringement of geographical indications of agricultural products in judicial cases is an effective way to deter the infringers with both rationality and low cost. In the current cases of the infringement of geographical indications of agricultural products, the infringement liability, especially the amount of damages, is extremely light. The two main reasons for this phenomenon are as follows: firstly, the plaintiff has difficulty in adducing evidence; secondly, the court has insufficient ability to consider the amount of damages. In the process of adducing evidence, it is usually difficult for the plaintiff to prove the actual loss he suffered as a result of the defendant’s tortious act, as well as the profit the defendant obtained as a result of the tortious act, and it is more difficult to adduce evidence on the more abstract reputational damage. Nevertheless, what is more important in determining the defendant’s tort liability is the court’s ability to consider the liability. In the first-instance case of the trademark infringement dispute between Hangzhou West Lake District Longjing Tea Industry Association and Putian City Chengxiang District Yuanjishicai Tea Store (See Fujian Putian Intermediate People’s Court (2020) Min 03 Min Chu No. 160 Civil Judgment), neither the plaintiff nor the defendant provided sufficient evidence to prove the losses suffered due to the infringement or the amount of profits obtained due to the infringement. The court, when determining the amount of compensation to be borne by the defendant by applying the statutory compensation method, specifically considered the following factors, proving its ability to investigate the truth and assume a higher responsibility: Yuanjishicai Tea Store’s infringement was reflected in the title of the commodity and the details of the commodity in the commodity link, introducing and using the trademark involved in the case; Yuan’s total sales volume of 661 alleged infringing commodity links, and commodity links using “West Lake Longjing” sold by Caicha Tea Store reached approximately RMB 5760. The trademark involved in the plaintiff’s case was once recognized as a well-known trademark with high popularity, the plaintiff needed to pay corresponding reasonable expenses to stop the infringement.

Therefore, in the face of the dilemma that the amount of compensation for infringement damages is abnormally light, the judicial level should first standardize the legal compensation method. The fourth compensation method of trademark law, as a general clause, is widely applied in cases of the infringement of geographical indications of agricultural products, which has become the inevitable criterion for the court to determine the amount of compensation for infringement. In order to alleviate this phenomenon, active judicature and strict judicature are eclectic. It is necessary to encourage judges to break the shackles of damages, choose the compensation method that is most beneficial to the obligee, refine the consideration basis of statutory compensation, and reasonably limit the discretion of judges. For example, in cases of the infringement of geographical indications of agricultural products that are subject to statutory compensation, judges need to classify the infringers, such as individual industrial and commercial households, small supermarkets, large supermarket chains, etc., and determine the minimum compensation amount according to the infringer’s business scale, organizational form, and local economic level, which is not only conducive to safeguarding the rights holder’s interests in damages, but also takes into account the infringer’s actual performance ability, so that the judgment will not end up as a dead letter. The court of Xinyang Tea Association and Foshan Shunde Baijiang Lifestyle Supermarket Co., Ltd. (Foshan, China), in the first instance of civil cases of trademark infringement disputes (See Guangdong Foshan Chancheng People’s Court (2021) Yue 0604 Minchu No. 4195 Civil Judgment), also considered the fact that “the defendant belongs to a small department store” and made an appropriate compensation judgment. Secondly, the amount of compensation for damages should reasonably increase. Nowadays, the phenomenon of infringement and counterfeiting of geographical indications of agricultural products emerges one after another, and the number of cases of infringement of geographical indications of agricultural products is very small compared with the actual number of infringements. The main reason is that the amount of compensation recognized by the court in the current judicial precedents is usually far below the expectations of the obligee, and it is not even enough to make up for the litigation cost, which seriously hits the obligee’s enthusiasm for prosecution. Additionally, when the amount of compensation for damages is lower than the profit that can be obtained from the infringement, the infringer will continue to choose the same way to make profits, which leads to the unhealthy trend of getting something for nothing in the market. Therefore, it is very important to reasonably increase the amount of compensation for damages and change the chilling phenomenon that the obligee is tired of defending rights and that the infringer ignores the infringement cost. Finally, punitive damages should be reasonably applied in cases of infringement of geographical indications of agricultural products. Since the introduction of punitive damages into the field of intellectual property, judges are still cautious, and few of them use it in judicial practice, and the impacts on disputes over infringement of geographical indications of agricultural products are even more ineffective. One of the reasons is that, although the Civil Code provides for the reasonable application of punitive damages in intellectual property cases in principle, there is no separate law for applying punitive damages to GIs at present, as other intellectual property objects usually have separate laws for punitive damages. Because the protection of geographical indications of agricultural products is similar to that of trademarks, in the absence of specific laws, the application of general rules of punitive damages in trademark law in cases of infringement of geographical indications of agricultural products does not exceed public expectations [52], and it also has the legal effect of combating infringement.

### 5.5. Control in the Process: Strengthening the Standardization Construction in the GI Industry of Agricultural Products with Quality Monitoring Mechanism as the Core

The quality and safety standards of geographical indications of agricultural products are not only related to the identification of their infringement but also to social food security. In this regard, the European Union provides a regulatory framework and official monitoring procedures to ensure the product production conditions required for the effective implementation of GIs are met [53]. There are still some shortcomings and hidden problems in the construction of the standardization of geographical indications in the agricultural industry in China. Therefore, it is necessary to focus on the quality monitoring mechanism and speed up the construction of the whole industrial chain’s standard system of geographical indications of agricultural products.

The key to the effective operation of the quality monitoring mechanism lies in the realization of “one net” of supervision and the formation of a supervision grid that integrates the upper and lower parts and cooperates with the left and right parts. It is necessary to strengthen the key role of the government of origin of GI products in the implementation of GI standards, to regularly monitor and evaluate the implementation effect of standards, and to strictly control the quality level of all links of geographical indications of agricultural products. The main responsibilities of the five levels of city, county, township, village, and enterprise, which, respectively, correspond to the main responsibilities of risk monitoring and analysis, supervision and spot check, complaint reporting and handling, management of quality supervisors, household survey and reporting of risks, and control and tracing of key links of internal inspectors [54], should be clarified. In addition, the whole industrial chain standard of geographical indications of agricultural products is the basis for the operation of the quality monitoring mechanism, and it is necessary to thoroughly implement the spirit of the 14th Five-Year Plan for the Protection and Application of GIs. With the participation of local intellectual property management departments, our protection department is responsible for speeding up the formulation and revision of national standards for geographical indications of agricultural products; speeding up the construction of a standard system for the whole industrial chain; covering the protection, application, management, and service of GIs; and implementing the requirements of “standard implementation, standard formulation when there is no standard, and standard supplement when lacking standard”. In the standardization construction of the whole industry chain, especially in the quality standard construction of geographical indications of agricultural products, it is necessary to ensure the characteristics and quality of geographical indications of agricultural products so that the products cannot be confused with ordinary products, which can be easily reproduced by ordinary products. Therefore, in the process of jointly negotiating, defining, and standardizing the implementation of standard rules, manufacturing groups gain and maintain product reputation and listing opportunities and avoid “hitchhiking” [55].

To strengthen the standardization construction of geographical indications of the agricultural products industry, it is also necessary to deal with disputes about the use of place names of common similar commodities in the process of geographical indications of agricultural products’ protection. In reality, place names commonly used on the packaging of commodities are recognized by consumers as places of origin or GIs, which often enjoy high popularity. Therefore, when place names are generally marked on the packaging of commodities, their recognition as GIs is far greater than that of place names. Whether it is the need of market management or of judicial adjudication, it is urgent to solve the problem that the standards of proper use of place names are not unified. The most direct way is for competent departments such as the General Administration of Market Supervision and Administration to issue documents to standardize the labeling methods of the origin of similar products using geographical indications of agricultural products, including labeling of the location of the origin, the obligation to disclose that the products are not GI products, etc., so as to better link the practice with the GI protection system. In the Administrative Measures for the Use of Special Signs of GIs (for Trial Implementation), although it is stipulated that “a unified social credit code should be marked at the designated position of the special signs of GIs”, the identification degree of geographical indications of agricultural products relative to common similar commodities is increased, but it cannot completely rule out the possibility that consumers will fall into misunderstanding. Standardizing the proper use of place names of common similar commodities can be used as the second barrier for the protection of geographical indications of agricultural products, which is of great significance to the operation of quality supervision mechanisms, the advancement of industrial standards for geographical indications of agricultural products, and the fight against malicious infringement of geographical indications of agricultural products.

## 6. Conclusions

Using a quasi-case research method and typological thinking, this paper systematically presents the judicial profile of the infringement dispute of geographical indications of agricultural products in China from five dimensions: the distribution of types of plaintiffs, the distribution of types of defendants, the distribution of types of infringement, the distribution of bases for the adjudication, and the distribution of criteria for adjudication and compensation. On this basis, the authors consider the dimensions of the infringement and conclude with the judicial deviation of the infringement of geographical indications of agricultural products. The research results show that the infringement of geographical indications of agricultural products deviates between the implicitness of infringement behavior and the explicitness of the object of infringement; deviates between the expectation of the implementation of the behavior and the lag of the introduction of geographical indications of agricultural products; deviates between the concreteness of the behavior and the abstraction of the identification standard of infringement; and deviates between the clarity of the qualitative standard and the fuzziness of the quantitative standard, which reflects the necessity of deviation control of the infringement of geographical indications of agricultural products. In this paper, the focus of the controversy in judicial regulation practice is reflected. In view of the disputes over the identification of infringement, the use of geographical names, and the liability for infringement of geographical indications of agricultural products, based on the reality of China and absorbing excellent international experience, we created a regulatory path for the infringement of geographical indications of agricultural products with a Chinese style and characteristics, including introducing procuratorial public interest litigation into the protection of geographical indications of agricultural products, realizing the full coverage of the infringement of geographical indications of agricultural products through multi-agent cooperation, reasonably confirming the amount of compensation for the infringement of geographical indications of agricultural products, and strengthening the standardization of geographical indications of agricultural products, with a quality monitoring mechanism at the core.

The conclusion of the manuscript enlightens the construction of and improvement in the protection system of geographical indications of agricultural products in China and even other countries around the world. Previous studies only focused on the one-way protection and trademark rights of geographical indications of agricultural products by law, but this study, based on the existing one-way protection provisions of laws and policies, focused on the “infringement” itself, starting from the essential attribute of infringement; innovated the protection measures of geographical indications of agricultural products in the aspects of industrial management, infringement relief, and standardization construction, so as to realize the multi-directional regulation and whole-process regulation of geographical indications of agricultural products infringement; and strengthened the all-round protection of geographical indications of agricultural products.

## Figures and Tables

**Figure 1 ijerph-20-04946-f001:**
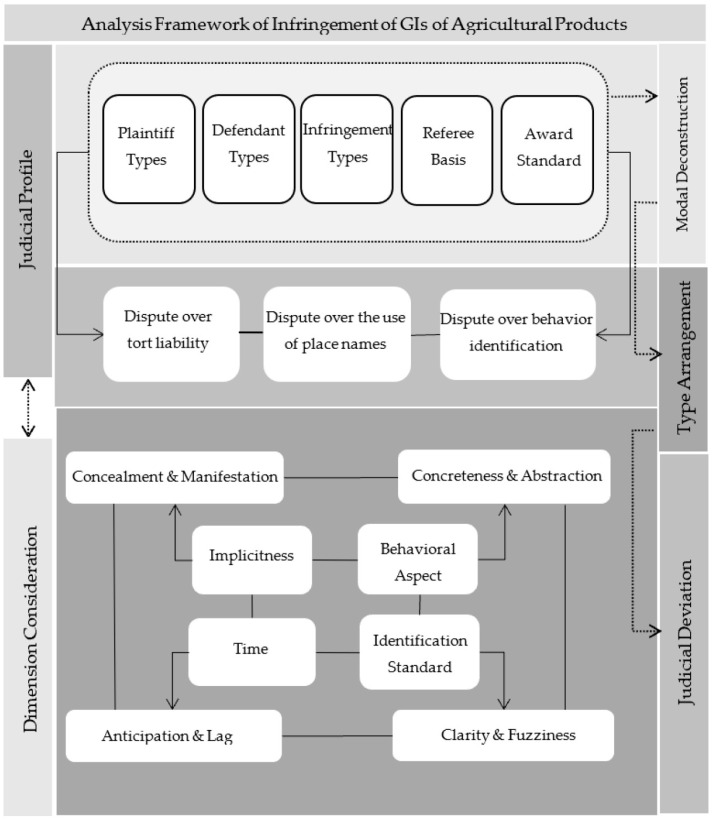
Analysis Framework of Infringement of Geographical Indications of Agricultural Products.

**Figure 2 ijerph-20-04946-f002:**
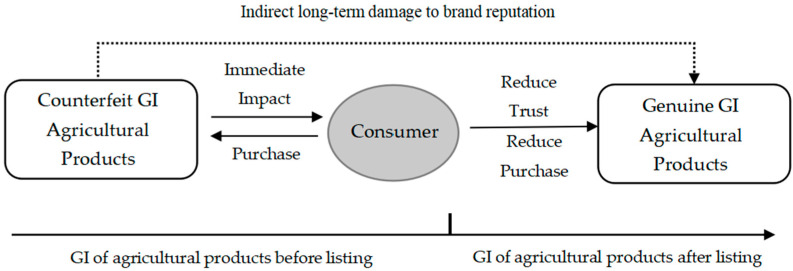
Reaction chain of bad money driving out good money as caused by infringement.

**Figure 3 ijerph-20-04946-f003:**
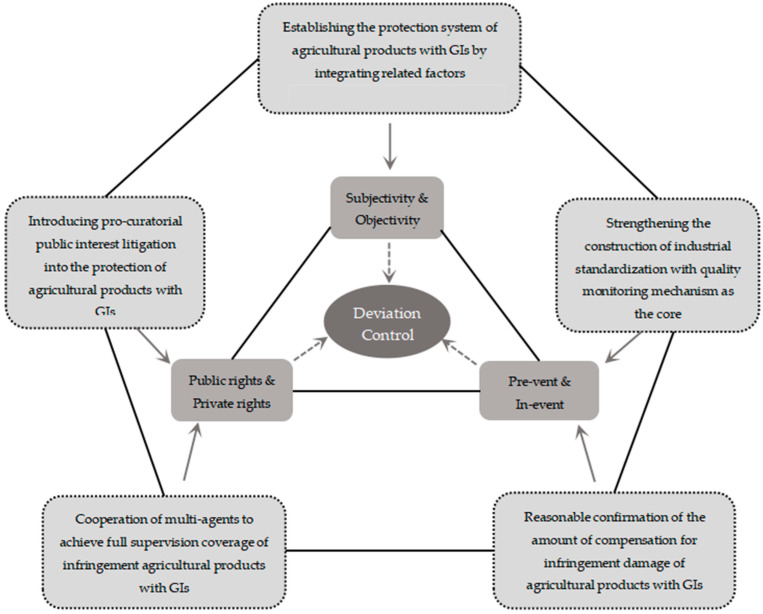
Regulatory framework for infringement of geographical indications of agricultural products.

**Table 1 ijerph-20-04946-t001:** Existing literature on the GIs of agricultural products.

Topic	Perspective	Research Site	Method	Publication
Protection of geographical indications (GIs)	Subject participation; Governance model; Institutional evolution; Practical defects; Collective action; Quality standard	Developed country	Quantitative analysis	[41]
				[34]
			Comparative case study approach	[43]
				[38]
				[37]
			Qualitative method	[46]
				[45]
		Developed country and developing country	Qualitative method	[44]
		Developing country	Qualitative method	[35]
				[33]
				[31]
		None	Qualitative method	[36]
Attributes and functions of GIs	Intellectual property; Rural vitalization; Global value chain	Developed country	Case study	[21]
			Empirical research	[23]
				[20]
				[19]
			Qualitative method	[28]
		Developing country	Qualitative method	[27]
				[26]
			Empirical analysis	[25]
		None	Empirical analysis	[24]
			Case study	[22]
			Literature review	[42]
			Quantitative analysis	[32]
Infringement of GIs	Judicial investigation	Developing country	Quasi-case research	[40]
	Essence of infringement of GIs	Developing country	Quasi-case research	Manuscript
**Advantages of this manuscript**				
The existing literature lacks attention to the infringement of GIs of agricultural products in developing countries in sample selection, fails to focus on “infringement” in the research object, and fails to use judicial cases to summarize in terms of the research methods. The uniqueness of this article lies in using China, a representative developing country, as an example and taking “the infringement of Gis of agricultural products” as the research object. It systematically sorts out the judicial patterns of the disputes over the infringement of Gis of agricultural products in China, summarizes the main points of contention after sorting out the types, explores and analyzes the judicial profile of the infringement and the overall outline in practice, considers the dimensions of the infringement of Gis of agricultural products, and digs out the characteristics of the infringement, so as to clarify the judicial deviation of the infringement of Gis of agricultural products.				

**Table 2 ijerph-20-04946-t002:** Case list in manuscript content.

Judicial Documents Number	Focus of Controversy
(2018) Zhe 11 Minchu No. 84	1. Whether the defendant, Hongming Tea Company, infringed on the plaintiff’s trademark right and constituted unfair competition; 2. Whether the defendant, Alibaba Company, as a service provider of an online trading platform, should bear civil liability for the alleged infringement of the defendant, Hongming Tea Company; 3. If it constitutes infringement, what kind of civil liability the defendant should bear.
(2021) Jing 0102 Minchu No. 16099	1. Whether the defendant, Hefei Huixuan Tea Co., Ltd. (Hefei, China), infringed onthe trademark right of Xinyang Tea Association and constituted unfair competition; 2. If it constitutes infringement, what kind of civil liability the defendant should bear.
(2017) Jin 01 Minchu No. 405	1. Whether the defendant, Chen Qijun Seasoning Wholesale Center, in Xiqing District of Tianjin, infringed on the trademark right of Panjin Rice Association and constituted unfair competition; 2. If it constitutes infringement, what kind of civil liability the defendant should bear.
(2022) Xiang 0104 Minchu No. 5266	Whether the defendant is responsible for infringement and is liable for compensation.
(2020) Yue 0106 Minchu No. 12496	Whether the defendant is responsible for infringement and is liable for compensation.
(2015) Yue Zhi Minzhong No. 202	Whether the defendant is responsible for infringement and is liable for compensation.
(2020) Yue 73 Minzhong No. 4846	Whether the logo used by Li Minfu Company (Xinyang China) on the alleged infringing products infringes on the exclusive right to use registered trademarks of Xiantao Rice Industry Association.
(2022) Xiang 0104 Minchu No. 5266	Whether the defendant is responsible for infringement and is liable for compensation.
(2015) Jing Zhi Minzhong No. 1180	1. Whether the position of Jinli Xingsheng Company in the alleged infringement and the relevant determination of the first-instance judgment are wrong; 2. Whether the amount of compensation determined by the first-instance judgment and the reasonable expenses of litigation are improper.
(2020) Qing 01 Zhi Minchu No. 38	1. Whether the sale of the apples involved by the Song Fruit Firm infringes on the exclusive right to use the trademark of the Apple Association in Aksu; 2. Whether the civil lawsuit filed by the Apple Association in Aksu region complies with the law.
(2021) Qing Zhi Minzhong No. 20	1. Whether the apples involved in the case sold by Xingmin Commercial Bank are from Wensu County in Aksu region; 2. Whether its sales of the apples involved infringed on the exclusive right to use trademarks of the Apple Association in Aksu.
(2016) Hu 0110 Minchu No. 3624	1. Whether the defendant infringed on the plaintiff’s exclusive right to use the registered trademark involved; 2. The kind of legal responsibility the defendant should bear.
(2017) Liao 12 Minchu No. 94	1. Whether the defendant infringed on the plaintiff’s exclusive right to use the registered trademark involved; 2. What kind of legal responsibility the defendant should bear.
(2017) Jin 01 Minchu No. 404	1. Whether the defendant infringed on the plaintiff’s exclusive right to use the registered trademark involved; 2. What kind of legal responsibility the defendant should bear.
(2022) Hu 0104 Minchu No. 12355	1. Whether the defendant infringed on infringed on the plaintiff’s exclusive right to use the registered trademark involved; 2. What kind of legal responsibility the defendant should bear?
(2020) Zhe 0382 Minchu No. 5440	1. Whether the defendant infringed on the plaintiff’s exclusive right to use the registered trademark involved; 2. What kind of legal responsibility the defendant should bear.
(2021) Yue 2071 Minchu No. 11902	1. The fact that Shunhe Department Store raised the issue of legitimate source defense. 2. The amount of compensation.
(2020) Min 03 Minchu No. 160	1. Whether the defendant infringed on the plaintiff’s exclusive right to use the registered trademark involved; 2.What kind of legal responsibility the defendant should bear.
(2021) Yue 0604 Minchu No. 4195	1. Whether the defendant infringed on the plaintiff’s exclusive right to use the registered trademark involved; 2. What kind of legal responsibility the defendant should bear.

Note: Judicial documents number comes from PKULAW Network: https://www.pkulaw.com/.

**Table 3 ijerph-20-04946-t003:** Distribution of prosecution subject structure.

Types of Trademark Owners	Certified Trademark Registrant				Certificate of Trademark User
**Type of Plaintiff**	Industry Association	Agricultural Bureau Industrial Station	Agricultural Technology Extension Center	Office of Industry Management Committee	Limited Liability Company
**Documents (copies)/Ratio**	198/ 97.06%	2/ 0.98%	2/ 0.98%	1/ 0.49%	1/ 0.49%

**Table 4 ijerph-20-04946-t004:** Type and structure of the sued subject.

Number of Copies/ Types	Documents (Copies)/ Ratio							
**Types of Sued Subjects**	Self-employed Businesses/ Self-employed Businesses and Self-employed Businesses	Limited Company/ Limited Company and Limited Company	Self-employed Businesses and Limited Companies	Professional Cooperatives and Limited Companies	**Types of Online Tort-Accused Subjects**	Separate Defendant	Co-defendant of Network Provider	Other Co-defendants
	139/ 68.13%	51/ 25%	11/ 5.37%	3/ 1.5%		5/ 19.23%	18/ 69.23%	3/ 11.54%

**Table 5 ijerph-20-04946-t005:** Distribution of infringement types.

Types of Infringement	Use Forged Types	Non-Standard Specification Types
**Documents(copies)/** **Ratio**	189/ 94.03%	12/ 5.97%

**Table 6 ijerph-20-04946-t006:** The application of law.

Clause	Article 57 of the Trademark Law	Article 16 of the Trademark Law	Article 3 of the Trademark Law	Article 48 of the Trademark Law
**Quantity**	198	163	157	123

**Table 7 ijerph-20-04946-t007:** Type distribution of first-trial compensation standard.

Type of Compensation Standard	Consider Only Rule Factors	Consider Both Rule Factors and Individual Factors
**Documents (copies)/** **Ratio**	168/ 83.58%	33/ 16.42%

**Table 8 ijerph-20-04946-t008:** Criteria for Determining Infringement and Presentation Table of Evidentiary Effect.

Judicial Documents Number	Evidence of Infringement Can/Cannot Be Identified	Evidentiary Force
(2022) Wan Minzhong No. 362	The evidence provided by Anji Tea Station can prove that the boxed tea in question was purchased by Anji Tea Station from Wuzhong Green Tea Company and that Wuzhong Green Tea Company used the registered trademark No. 14982232 “Anji White Tea” on the outer packaging of the tea sold without the permission of Anji Tea Station, the trademark registrant, which did not fall within the scope of fair use of GI certification trademarks, and according to relevant laws and regulations, the behavior of Wuzhong Green Tea Company constituted infringement.	effective
(2021) Su 12 Minzhong No. 3787	“Xinghua hairy crab” is a legally registered collective trademark and agricultural product with GIs, and the appellant failed to perform the legal process of using the collective trademark, nor did it perform the legal use process in accordance with the regulations on the administration of agricultural products with GIs, which is not a legal use.	effective
(2020) Su 08 Minzhong No. 3135	“Huaihei Eighteen Halogen” is a registered trademark obtained by Qiuge Company, but it has not obtained the famous trademark certification of Jiangsu Province and the certification of agricultural products with GIs, but Huai’an Black Pig has obtained the certification of geographical indication of agricultural products, and the infringement of Qiuge Company is established.	effective
(2020) Jing Xingzhong No. 6875	The evidence in the case can only prove that the Administrative Rules of the Mizhi Millet Association only deal with the general provisions that goods using the disputed trademark should “ensure the stable quality of products (or services)”, which is not clear or specific and lacks operability, and it does not meet the substantive requirements stipulated in the Administrative Measures.	void
(2020) Jing 73 Xingchu No. 730	The evidence submitted by the plaintiff at the litigation stage, the Certificate of Registration of Agricultural Products with GIs, was not the natural basis for obtaining the registration of the GI trademark, and the registration applicant shown in the registration certificate was Fuzhou Linchuan Jinshan Edible Mushroom Professional Cooperative, and it could not be determined that the relevant GI goods approved in the registration certificate were the trademark goods applied for in this case.	void
(2014) Wuhou Minchu No. 297	The registration scope of agricultural products with GIs is primary agricultural products derived from agriculture, so “Qingchuan wild tianma” is a primary agricultural product. According to the Law of the People’s Republic of China on the Quality and Safety of Agricultural Products, primary agricultural products are not included in the scope of food production licenses, and the defendant’s sale of “Tianma” as a primary agricultural product does not violate the relevant regulations of the Ministry of Health saying that “Tianma” cannot be produced or used as ordinary food raw materials.	void

Note: Judicial documents number comes from PKULAW Network: https://www.pkulaw.com/.

**Table 9 ijerph-20-04946-t009:** Table of Cases Presented in Dispute over the Use of Geographical Names.

Judicial Documents Number	The Case Involved Production Areas and Agricultural Products	Final Referee Results
(2022) Lu 0991 Minchu No. 958	The chili pepper produced in Zhangshugang is a specialty of Zhangshu Town, Xiangyin County, Yueyang City, Hunan Province, and is famous for its long history, specific production methods, and unique taste.	The defendant could neither prove that the chili pepper it sold met the conditions for using the GI, nor could it state the legal source of the chili pepper, so the defendant’s conduct constituted trademark infringement.
(2019) Jing 73 Minzhong No. 2601	The Hetao refers to the land east of Helan Mountain, west of Lüliang Mountain, south of Yin Mountain, and north of the Great Wall, and is an important wheat-producing area in China.	The trademark infringement of COFCO Haiyou Company and Qingdao Xinghua Company and the unfair competition behavior of COFCO Haiyou Company shall bear legal liabilities such as stopping the infringement, compensating for losses, and eliminating the impact.
(2018) Lu 09 Minchu No. 311	The ‘Golden Pigeon Mountain Millet’ declared by the Millet Association of Longgang Town, Linqu County, passed the review and the expert review organized by the Agricultural Products Quality and Safety Center of the Ministry of Agriculture, and implemented the registration and protection of national agricultural GIs.	Shandong Agricultural University also provided Yaoshan Cooperative with relevant regulations and evaluation reference materials on the application of national GI products of millet produced in the area by email, and Yaoshan Cooperative believes that the millet produced by it has been included in the national agricultural geographical indication protection products, so the court presumes that Shandong Agricultural University has completed the first type of content agreed upon by both parties.
(2018) Su Minzhong No. 891	As a high-end rice variety, “rice flower fragrance” rice has high market popularity and reputation and has become an important symbol for consumers to choose high-quality rice.	Hongxinyuan Company, as the producer of the allegedly infringing product, and Yipin Tangshan Supermarket, as the seller, shall bear the civil liability to stop the infringement and compensate for losses.
(2018) Gan 0271 Minchu No. 1313	Golden frog rice is produced in Wanchang, Jilin, a famous high-quality-rice-producing area in Northeast China. The area has an excellent ecological environment and is one of the 14 green industrial science and technology parks in China certified by the Green Expert Committee of the United Nations Industrial Development Organization	Jilin Yufeng Rice Industry Co., Ltd. (Baicheng, China) paid compensation of RMB 8670 to Hu Enguang within ten days after the effective date of this judgment.

Note: Judicial documents number comes from PKULAW Network: https://www.pkulaw.com/.

**Table 10 ijerph-20-04946-t010:** Table of Excessive Damages for Tort Liability.

Judicial Documents Number	Amount and Basis for Damages
(2022) Anhui Min Zhong No. 362	Taking into account factors such as the nature, period, and consequences of the infringement in this case, the reputation of the trademark involved in the case, and the cost of rights protection, it was determined that the company would compensate Qingcha in Wuzhong for economic losses of RMB 20,000, including reasonable expenses incurred to stop the infringement.
(2022) Lu 0783 Minchu No. 6484	In view of the fact that neither party in this case submitted sufficient evidence to prove the actual losses suffered by the plaintiff as a result of the infringement and the benefits obtained by the defendant as a result of the infringement, the circumstances of the infringement in question, the popularity of the trademark in question, the nature of the defendant, subjective fault, the duration of the sale, the price, and the reasonable expenses incurred by the plaintiff to stop the infringement were based on factors such as the circumstances of the infringement in question, the popularity of the trademark in question, the nature of the defendant, subjective fault, the duration of the sale, the price, and the reasonable expenses incurred by the plaintiff to stop the infringement. As appropriate, it was determined that the defendant compensated the plaintiff for economic losses (including reasonable expenses) totaling RMB 8500.
(2021) Su 12 Min Zhong No. 3787	Xinghua Hairy Crab Association incurred reasonable expenses such as investigating and collecting evidence to stop the infringement, entrusting lawyers to participate in litigation, and other factors. It was determined that the defendant Zhenxian Zi Company and the defendant Hehengshengde Aquaculture Society jointly compensated the Xinghua Hairy Crab Association for economic losses and reasonable expenses incurred for rights protection totaling RMB 100,000.
(2021) Yu 01 Minchu No. 7229	As for the specific amount of compensation, since the plaintiff did not provide evidence to prove the amount of its economic losses and the amount of profits made by the defendant, it was determined that the defendant must compensate the plaintiff for economic losses and reasonable expenses of RMB 50,000 according to the high popularity of the trademark involved in the case, the nature, period, business location, business scale, and the degree of fault of the defendant’s counterfeiting infringement, as well as the fact that the plaintiff hired a lawyer to appear in court and collect evidence on the spot for the litigation in this case and needed to incur corresponding reasonable rights protection costs.
(2020) Su 02 Minchu No. 481	Taking into account the popularity of the registered trademark involved in the case, the business scale of Yunjue Company, the sales price, the sales quantity, the sales mode of the infringing packaging materials involved in the case, the subjective fault of Yunjue Company, the reasonable expenses incurred by the Yangshan Peach Farmers Association to stop the infringement, and other relevant factors, Yunjue Company was to compensate the Yangshan Peach Farmers Association for economic losses of RMB 120,000.
(2018) Su Min Zhong No. 891	It was determined that Hongxinyuan Company shall compensate RMB 100,000 for economic losses, and Yipin Tangshan Supermarket shall bear joint and several compensation for RMB 10,000. The notary fee of RMB 2000 and the cost of purchasing infringing goods of RMB 324.9 paid by Fuzhou Rice Mill for this case, a total of RMB 2324.9, were reasonable expenses paid by it to stop the infringement, and were to be compensated by Hongxinyuan Company, of which Yipin Tangshan Supermarket accounted for 1131.3%. The yuan is jointly and severally liable.

Note: Judicial documents number comes from PKULAW Network: https://www.pkulaw.com/.

**Table 11 ijerph-20-04946-t011:** Distribution of Request Amount and Judgment Odds of First-instance Cases in West Lake Longjing.

Request Amount and Judgment Odds of West Lake Longjing First-Instance Case						
**Type**	Maximum Requested Amount	Minimum Requested Amount		Maximum Compensation Ratio		Minimum Compensation Ratio
**First Instance Case**	400,000 (RMB)	50,000 (RMB)		90.9%		3%
**Interval Distribution of Judgment Odds in West Lake Longjing First-Instance Cases**						
**Compensation Ratio**	Less than 10%	10–19%	20–29%	30–39%	40–49%	50% and above
**Documents (copies)**	1	4	5	10	1	2

**Table 12 ijerph-20-04946-t012:** Distribution of infringement cases of Korla fragrant pear.

District	Henan	Beijing	Shandong	Jiangsu	Sichuan	Zhejiang	Heilongjiang	Shanghai	Guangdong
**Quantity**	12	4	3	3	3	1	1	1	1

## Data Availability

The data presented in this study are available on request from the corresponding author.

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
