# Peer review of "How to Regulate the Infringements of Geographical Indications of Agricultural Products—An Empirical Study on Judicial Documents in China"

_ijerph, 2023, doi:10.3390/ijerph20064946_

Round 1

Reviewer 1 Report (Previous Reviewer 1)

I agree with the authors' answers to my comments. The additions and given facts left me with no doubts about the originality of the research.

Everything is correctly completed.

Author Response

Dear Reviewer,

On behalf of my co-authors, were truly grateful for having this opportunity to revise our manuscript. I would like to express our sincere gratitude to the editors and reviewers for their constructive comments and suggestions on  our manuscript entitled How to regulate the infringements of geographical indications of agricultural products?——An empirical study on judicial documents in China . We have made revisions to the manuscript and have completed a point-by-point response to the comments. The revised content has been highlighted in red within the manuscript. Reviewer #1 approved our manuscript in the review comments, and did not provide any modification suggestions, therefore we only replied to reviewer #2' s suggestions in the point-by-point response. Please find a highlighted version of our revised paper and a point-by-point response version which We have uploaded in the submission system and would like to submit for your kind consideration.

Responds to Reviewer #2’s Comments Point by Point

Comment 1: Rewrite the abstract and summarize the problem as we need to know where you research conducted? If there is any sampling or not? Then methodology should be explained as you did some. So what was the result?

Response: We appreciate the suggestions from the reviewer#2 regarding the abstract section, which has greatly helped to enhance the presentation of the manuscript as well as the conciseness of the abstract.According to the reviewer #2's suggestion, we have revised and rewritten the abstract section. We re-summarized the contents of the manuscript from the aspects of research purpose, research method, sample selection,research results, and so on.Then we re-wrote it according to the usual format for abstracts in articles published in International Journal of Environmental Research and Public Health..

The specific added contents are as follows: 

Under the background of China's strategy of becoming a powerful agricultural country, geographical indications(GIs) of agricultural products, as an important intellectual property right to enable China-featured agriculture to develop with high quality, have a strong effect of strengthening and promoting agriculture. However, there are a large number of infringements of GIs of agricultural products in judicial practice, which not only greatly damage the economic and social values of GIs of agricultural products, but also bring huge food safety hazards to consumers and hinder the overall protection of intellectual property rights in China. Based on this, this paper, with the help of Quasi-case research method, integrates the facts of relevant cases, the focus of disputes, the application of law and other case elements to realize the case similarity judgment based on the legal argumentation model. With the help of the retrieval tool of "Peking University Magic Weapon", this paper makes statistics on the civil cases of infringement of GIs of agricultural products in China from January 1, 2014 to July 31, 2022, and sets different retrieval conditions for two searches. After twice screening, 245 valid samples were obtained, and the judicial patterns of infringement disputes over GIs of agricultural products in China were systematically sorted out from the distribution of plaintiff and defendant, the distribution of infringement types, the basis of adjudication and the standard of compensation. It was found that the plaintiff types showed double simplification, the infringement types took the edge infringement as the basic form, and the general trademark provisions occupied the main position in legal application. Then, the main litigation points such as the dispute over the identification of GIs of agricultural products, the dispute over the use of geographical names and the dispute over tort liability are summarized, so as to dig out the characteristics of the implicitness of infringement, the expectation of implementation and the concreteness of aspects. On this basis, the regulatory path of the infringement of GIs of agricultural products is put forward, such as introducing procuratorial public interest litigation, multi-agents cooperating to implement all-round supervision, and reasonably determining the amount of damages.

Comment 2: Please revise and shorten following sentence: “Based on this, with Quasi-case research methods and typological thinking, this paper systematically sorts out the judicial patterns of infringement disputes over GIs of agricultural products in China from the distribution of plaintiff and defendant, the distribution of infringement types, the basis of adjudication and the standard of compensation. Also the paper sums up the main litigation points including the dispute over the identification of GIs of agricultural products, the use of geographical names and tort liability for infringement after sorting out the types, so as to explore and analyse the judicial profile of infringement and the overall outline in practice in China”.  

In page 3, you mentioned three questions. Please rewrite the first one as it is not questioning.

Response: We have reorganized this passage according to the reviewer #2's suggestion, focusing on the analytical framework of the article from sample deconstruction to type sorting to dispute induction and feature summary.At the same time,We have also made modifications to the three questions raised to ensure they meet the requirements for academic expression. 

The specific added contents are as follows: 

Based on this, this paper takes "the infringement of GIs of agricultural products" as the research object, systematically sorts out the judicial patterns of the disputes over the infringement of GIs of agricultural products in China, sums up the main points of litigation after sorting out the types, explores the judicial profile of the infringement and the overall outline in practice, and considers the dimensions of the infringement of GIs of agricultural products to dig out the characteristics of the infringement, so as to clarify the judicial deviation of the infringement of GIs of agricultural products. The following three questions are replied: First, whether there are obvious laws in the judicial cases of infringement of GIs of agricultural products, such as the type of plaintiff, the respondent and the application of law; Second, what are the controversial points in the regulation of infringement of GIs of agricultural products; Third, how to reverse the judicial deviation of the infringement of GIs of agricultural products.

Comment 3: The following phrase repeated over 6 times in abstract, I think better you totally revise the abstract or use abbreviation: “GIs of agricultural products.”Geographical indication use all over the text as GI/GIs as you did in some parts.

Response: Thanks to the reviewer #2's consideration of the details of the article, we have checked the full text according to the reviewer's suggestion to ensure that all abbreviations about "GIs of agricultural products" are consistent.

Comment 4Add the structure of the paper at the end of section 1.

ResponseAccording to the reviewer's suggestion, we have added an article structure at the end of section1 to briefly summarize the contents of Sections 2, 3, 4, 5 and 6 respectively.

The specific added contents are as follows: 

The rest of the article is carried out according to the following structure: section 2 puts forward the shortcomings of the existing research system on the infringement of GIs of agricultural products and the innovation of this article on the basis of summarizing the existing literature; Section 3 introduces the samples selected in this paper and the Quasi-case research method adopted, and deconstructs the samples on this basis; Section 4 sorts out the types and summarizes the main points of contention on the basis of sample deconstruction, and considers the infringement of GIs of agricultural products in different dimensions; Section 5 puts forward effective measures to regulate the infringement of GIs of agricultural products; Section 6 restates the research significance of the article while summarizing the conclusion, and looks forward to the future research.

Comment 5 I think better you move figure 1 to section 3 as here in literature section you need to sum up the previous works in table, any criteria/ any characteristics/ method/ technique and…Also better add your work and its advantages at the end of this table.

Response: Thanks for the reviewer #2's overall consideration of the structure of the full text, we have moved Figure 1 to section 3 of the article, and added a table at the end of the reference to systematically sort out the relevant research on GIs of agricultural products in the existing literature, and summarized the research paths, methods and technical characteristics of the existing research, and at the same time presented the complement of this manuscript to the existing research.

The specific added contents are as follows: 

Table 1. Existing literature on the GIs of agricultural products

Topic

Perspective

Research Site

Method

Publication

Protection of GIs

Subject participation;

Governance model;

Institutional evolution;

Practical defects;

Collective action;

Quality standard

Developed countries

Quantitative analysis

[41]

[34]

Comparative case study approach

[43]

[38]

[37]

Qualitative method

[46]

[45]

Developed countries and developing country

Qualitative method

[44]

Developing country

Qualitative method

[47]

[35]

[33]

[31]

None

Qualitative method

[36]

Attributes and functions of GIs

Intellectual property;

Rural vitalization;

Global value chain

Developed countries

Case study

[21]

Empirical research

[23]

[20]

[19]

Qualitative method

[28]

Developing country

Qualitative method

[27]

[26]

Empirical analysis

[25]

None

Empirical analysis

[24]

Case study

[22]

Literature review

[42]

Quantitative analysis

[32]

Infringement of GIs

Judicial investigation

Developing country

Quasi-case research

[40]

Essence of infringement of GIs

Developing country

Quasi-case research

Manuscript

Advantages of the manuscript

The existing literature lacks attention to the infringement of GIs of agricultural products in developing countries in sample selection, fails to focus on "infringement" in the research object, and fails to use judicial cases to summarize in terms of the research methods. The uniqueness of this article lies in using China, a representative developing country, as an example and taking "the infringement of GIs of agricultural products" as the research object.It systematically sorts out the judicial patterns of the disputes over the infringement of GIs of agricultural products in China, summarizes the main points of contention after sorting out the types, explores and analyzes the judicial profile of the infringement and the overall outline in practice, considers the dimensions of the infringement of GIs of agricultural products, and digs out the characteristics of the infringement, so as to clarify the judicial deviation of the infringement of GIs of agricultural products.

We confirm that neither the manuscript nor any parts of its content are currently under consideration or published in another journal. All authors have approved the manuscript and agreed with its submission to International Journal of Environmental Research and Public Health.

We appreciate your consideration of our manuscript and look forward to your reply and suggestions. If you have any questions, please do not hesitate to contact us at the following address.

Sincerely

Lingling Li, Yingzi Chen, Haoran Gao and Changjian Li

7th March, 2023

Northwest A&F University,  7th Mar,2023

Reviewer 2 Report (New Reviewer)

-        Rewrite the abstract and summarize the problem as we need to know where you research conducted? If there is any sampling or not? Then methodology should be explained as you did some. So what was the result?

-        Please revise and shorten following sentence: “Based on this, with quasi-case research methods and typological thinking, this paper systematically sorts out the judicial patterns of infringement disputes over geographical indications of agricultural products in China from the distribution of plaintiff and defendant, the distribution of infringement types, the basis of adjudication and the standard of compensation. Also the paper sums up the main litigation points including the dispute over the identification of geographical indications of agricultural products, the use of geographical names and tort liability for infringement after sorting out the types, so as to explore and analyse the judicial profile of infringement and the overall outline in practice in China”.

-        The following phrase repeated over 6 times in abstract, I think better you totally revise the abstract or use abbreviation: “geographical indications of agricultural products.”

-        Geographical indication use all over the text as GI/GIs as you did in some parts.

-        In page 3, you mentioned three questions. Please rewrite the first one as it is not questioning.

-        Add the structure of the paper at the end of section 1.

-        I think better you move figure 1 to section 3 as here in literature section you need to sum up the previous works in table, any criteria/ any characteristics/ method/ technique and…Also better add your work and its advantages at the end of this table.

-        The rest of paper organized well.

Author Response

Dear Reviewer,

On behalf of my co-authors, were truly grateful for having this opportunity to revise our manuscript. I would like to express our sincere gratitude to the editors and reviewers for their constructive comments and suggestions on  our manuscript entitled How to regulate the infringements of geographical indications of agricultural products?——An empirical study on judicial documents in China . We have made revisions to the manuscript and have completed a point-by-point response to the comments. The revised content has been highlighted in red within the manuscript. Reviewer #1 approved our manuscript in the review comments, and did not provide any modification suggestions, therefore we only replied to reviewer #2' s suggestions in the point-by-point response. Please find a highlighted version of our revised paper and a point-by-point response version which We have uploaded in the submission system and would like to submit for your kind consideration.

Responds to Reviewer #2’s Comments Point by Point

Comment 1: Rewrite the abstract and summarize the problem as we need to know where you research conducted? If there is any sampling or not? Then methodology should be explained as you did some. So what was the result?

Response: We appreciate the suggestions from the reviewer#2 regarding the abstract section, which has greatly helped to enhance the presentation of the manuscript as well as the conciseness of the abstract.According to the reviewer #2's suggestion, we have revised and rewritten the abstract section. We re-summarized the contents of the manuscript from the aspects of research purpose, research method, sample selection,research results, and so on.Then we re-wrote it according to the usual format for abstracts in articles published in International Journal of Environmental Research and Public Health..

The specific added contents are as follows: 

Under the background of China's strategy of becoming a powerful agricultural country, geographical indications(GIs) of agricultural products, as an important intellectual property right to enable China-featured agriculture to develop with high quality, have a strong effect of strengthening and promoting agriculture. However, there are a large number of infringements of GIs of agricultural products in judicial practice, which not only greatly damage the economic and social values of GIs of agricultural products, but also bring huge food safety hazards to consumers and hinder the overall protection of intellectual property rights in China. Based on this, this paper, with the help of Quasi-case research method, integrates the facts of relevant cases, the focus of disputes, the application of law and other case elements to realize the case similarity judgment based on the legal argumentation model. With the help of the retrieval tool of "Peking University Magic Weapon", this paper makes statistics on the civil cases of infringement of GIs of agricultural products in China from January 1, 2014 to July 31, 2022, and sets different retrieval conditions for two searches. After twice screening, 245 valid samples were obtained, and the judicial patterns of infringement disputes over GIs of agricultural products in China were systematically sorted out from the distribution of plaintiff and defendant, the distribution of infringement types, the basis of adjudication and the standard of compensation. It was found that the plaintiff types showed double simplification, the infringement types took the edge infringement as the basic form, and the general trademark provisions occupied the main position in legal application. Then, the main litigation points such as the dispute over the identification of GIs of agricultural products, the dispute over the use of geographical names and the dispute over tort liability are summarized, so as to dig out the characteristics of the implicitness of infringement, the expectation of implementation and the concreteness of aspects. On this basis, the regulatory path of the infringement of GIs of agricultural products is put forward, such as introducing procuratorial public interest litigation, multi-agents cooperating to implement all-round supervision, and reasonably determining the amount of damages.

Comment 2: Please revise and shorten following sentence: “Based on this, with Quasi-case research methods and typological thinking, this paper systematically sorts out the judicial patterns of infringement disputes over GIs of agricultural products in China from the distribution of plaintiff and defendant, the distribution of infringement types, the basis of adjudication and the standard of compensation. Also the paper sums up the main litigation points including the dispute over the identification of GIs of agricultural products, the use of geographical names and tort liability for infringement after sorting out the types, so as to explore and analyse the judicial profile of infringement and the overall outline in practice in China”.  

In page 3, you mentioned three questions. Please rewrite the first one as it is not questioning.

Response: We have reorganized this passage according to the reviewer #2's suggestion, focusing on the analytical framework of the article from sample deconstruction to type sorting to dispute induction and feature summary.At the same time,We have also made modifications to the three questions raised to ensure they meet the requirements for academic expression. 

The specific added contents are as follows: 

Based on this, this paper takes "the infringement of GIs of agricultural products" as the research object, systematically sorts out the judicial patterns of the disputes over the infringement of GIs of agricultural products in China, sums up the main points of litigation after sorting out the types, explores the judicial profile of the infringement and the overall outline in practice, and considers the dimensions of the infringement of GIs of agricultural products to dig out the characteristics of the infringement, so as to clarify the judicial deviation of the infringement of GIs of agricultural products. The following three questions are replied: First, whether there are obvious laws in the judicial cases of infringement of GIs of agricultural products, such as the type of plaintiff, the respondent and the application of law; Second, what are the controversial points in the regulation of infringement of GIs of agricultural products; Third, how to reverse the judicial deviation of the infringement of GIs of agricultural products.

Comment 3: The following phrase repeated over 6 times in abstract, I think better you totally revise the abstract or use abbreviation: “GIs of agricultural products.”Geographical indication use all over the text as GI/GIs as you did in some parts.

Response: Thanks to the reviewer #2's consideration of the details of the article, we have checked the full text according to the reviewer's suggestion to ensure that all abbreviations about "GIs of agricultural products" are consistent.

Comment 4Add the structure of the paper at the end of section 1.

ResponseAccording to the reviewer's suggestion, we have added an article structure at the end of section1 to briefly summarize the contents of Sections 2, 3, 4, 5 and 6 respectively.

The specific added contents are as follows: 

The rest of the article is carried out according to the following structure: section 2 puts forward the shortcomings of the existing research system on the infringement of GIs of agricultural products and the innovation of this article on the basis of summarizing the existing literature; Section 3 introduces the samples selected in this paper and the Quasi-case research method adopted, and deconstructs the samples on this basis; Section 4 sorts out the types and summarizes the main points of contention on the basis of sample deconstruction, and considers the infringement of GIs of agricultural products in different dimensions; Section 5 puts forward effective measures to regulate the infringement of GIs of agricultural products; Section 6 restates the research significance of the article while summarizing the conclusion, and looks forward to the future research.

Comment 5 I think better you move figure 1 to section 3 as here in literature section you need to sum up the previous works in table, any criteria/ any characteristics/ method/ technique and…Also better add your work and its advantages at the end of this table.

Response: Thanks for the reviewer #2's overall consideration of the structure of the full text, we have moved Figure 1 to section 3 of the article, and added a table at the end of the reference to systematically sort out the relevant research on GIs of agricultural products in the existing literature, and summarized the research paths, methods and technical characteristics of the existing research, and at the same time presented the complement of this manuscript to the existing research.

The specific added contents are as follows: 

Table 1. Existing literature on the GIs of agricultural products

Topic

Perspective

Research Site

Method

Publication

Protection of GIs

Subject participation;

Governance model;

Institutional evolution;

Practical defects;

Collective action;

Quality standard

Developed countries

Quantitative analysis

[41]

[34]

Comparative case study approach

[43]

[38]

[37]

Qualitative method

[46]

[45]

Developed countries and developing country

Qualitative method

[44]

Developing country

Qualitative method

[47]

[35]

[33]

[31]

None

Qualitative method

[36]

Attributes and functions of GIs

Intellectual property;

Rural vitalization;

Global value chain

Developed countries

Case study

[21]

Empirical research

[23]

[20]

[19]

Qualitative method

[28]

Developing country

Qualitative method

[27]

[26]

Empirical analysis

[25]

None

Empirical analysis

[24]

Case study

[22]

Literature review

[42]

Quantitative analysis

[32]

Infringement of GIs

Judicial investigation

Developing country

Quasi-case research

[40]

Essence of infringement of GIs

Developing country

Quasi-case research

Manuscript

Advantages of the manuscript

The existing literature lacks attention to the infringement of GIs of agricultural products in developing countries in sample selection, fails to focus on "infringement" in the research object, and fails to use judicial cases to summarize in terms of the research methods. The uniqueness of this article lies in using China, a representative developing country, as an example and taking "the infringement of GIs of agricultural products" as the research object.It systematically sorts out the judicial patterns of the disputes over the infringement of GIs of agricultural products in China, summarizes the main points of contention after sorting out the types, explores and analyzes the judicial profile of the infringement and the overall outline in practice, considers the dimensions of the infringement of GIs of agricultural products, and digs out the characteristics of the infringement, so as to clarify the judicial deviation of the infringement of GIs of agricultural products.

We confirm that neither the manuscript nor any parts of its content are currently under consideration or published in another journal. All authors have approved the manuscript and agreed with its submission to International Journal of Environmental Research and Public Health.

We appreciate your consideration of our manuscript and look forward to your reply and suggestions. If you have any questions, please do not hesitate to contact us at the following address.

Sincerely

Lingling Li, Yingzi Chen, Haoran Gao and Changjian Li

7th March, 2023

Northwest A&F University,  7th Mar,2023

This manuscript is a resubmission of an earlier submission. The following is a list of the peer review reports and author responses from that submission.

Round 1

Reviewer 1 Report

Article 1

How to regulate the infringements of Agricultural products  with geographical indications——An empirical study on judicial documents in China

Lingling Li1, Yingzi Chen1 and Changjian Li2,*

The article does not present a single document as evidence of violations of the laws on the use of trademarks.

As a reviewer, this more amorphous idea of scientists is not confirmed by the facts. Only numbers are given.

The object of research is not presented in the article. It is necessary at least a few samples with a violation of the use of a trademark on agricultural products, for example, to show a court decision. Because when studying the research presented in the article, one gets the impression that the data are made up, the statements are far-fetched.

The general appearance of the article is not structured in accordance with the requirements of the magazine design Int. J. Environ. Res. Public Health

The research results section is not highlighted separately;

2. Methods;

3. Data Analysis

4. Results

5. Discussion

Reviewer 2 Report

The topic is super interesting and important, and the authors do make an effort to pass it along, but the English level is not appropriate for international publishing standards in law. This is actually regretful because again, the topic deserves attention and what the authors try to convey is meaningful (…though novelty should be spelled out more explicitly! Has the procuratorate’s civil public interest litigation never been proposed in literature? Not even in Mandarin?). I advise them to either publish directly in Mandarin, or to seek a co-author who is experienced with international law publishing in English and can help on the linguistic side. After that, the piece can be submitted to an English-language law journal specialised in IP or food policies. On the more substantial side, the authors may want to consider: 1) Literature on GI-associated know-how understood as trade secrets: because trade secrets are data assets that play a strategic role in the commercial policy of contemporary economies (see extensively, for Asia particularly: https://doi.org/10.5070/P8371048804), because Chinese lawmakers hold trade-secret protection in the highest regard, and because GIs are already partly protectable under the AUCL that also protects trade secrets it is worth exploring all interfaces between GIs as data know-how and its protection under China’s trade-secret legal regime. This would help in devising a general strategy on IP enforcement in China that can encompass both trade secrets and GIs by relying on common prosecutorial mechanisms, judicial assessments of damages/compensation, and industry incentives, leading to the formulation of policy recommendations and the identification of trends which are more generalizable beyond GIs per se. 2) For context, the author might wish to consult https://www.cambridge.org/core/books/geographical-indications-at-the-crossroads-of-trade-development-and-culture/unique-type-of-cocktail-protection-of-geographical-indications-in-china/B9D09B0EE88F1609945C59968B1C5FCA . 3) There are a few massive mistakes, for example you cannot write “countries such as the European Union”, at best you can write “organisations such as the European Union” or “countries such as the European Union’s Member States”. 4) In the introductory sections, and especially towards the literature review, I would cut all sections on how GIs contribute to agricultural and urban policing (we all know that), and rather introduce some solid literature on counterfeiting and IP enforcement in China, starting with https://www.journals.uchicago.edu/doi/10.1086/689230. 5) In line with other types of IP, it is perhaps worth looking for relevant SPC Judicial Interpretations and Guiding Cases that can help shape China’s regulatory environment when it comes to enforcing GIs. The authors mention one Interpretation at p 7 but fail to state why it is so important. “Interpretations” are generally understood as non-binding, while in fact they play a fundamental role within China’s legal sources! In any case, it seems to me that the protection of agricultural products as GIs rests at the intersection of IP but also environmental, intangible heritage, and know-how policies; this convergence bears obvious effects on how enforcement should work, and the authors at this stage fail to really bring that to surface.